# Clinical sequencing of soft tissue and bone sarcomas delineates diverse genomic landscapes and potential therapeutic targets

Benjamin A. Nacev [1,2,3,15], Francisco Sanchez-Vega [4,14,15], Shaleigh A. Smith [4,5], Cristina R. Antonescu [6], Evan Rosenbaum[1,2], Hongyu Shi[7], Cerise Tang[6,8], Nicholas D. Socci[5,9], Satshil Rana[6], Rodrigo Gularte-Mérida [4], Ahmet Zehir [6], Mrinal M. Gounder[1,2], Timothy G. Bowler[1], Anisha Luthra[5,10], Bhumika Jadeja [4], Azusa Okada[4], Jonathan A. Strong[4], Jake Stoller[4], Jason E. Chan [1], Ping Chi[1,2,10], Sandra P. D'Angelo [1,2], Mark A. Dickson[1,2], Ciara M. Kelly [1,2], Mary Louise Keohan[1,2], Sujana Movva[1,2], Katherine Thornton[1,2], Paul A. Meyers[11], Leonard H. Wexler [11], Emily K. Slotkin[11], Julia L. Glade Bender [11], Neerav N. Shukla[11], Martee L. Hensley[1,2], John H. Healey [4], Michael P. La Quaglia[4,11,12], Kaled M. Alektiar[13], Aimee M. Crago[4,12], Sam S. Yoon[4,12], Brian R. Untch[4,12], Sarah Chiang[6], Narasimhan P. Agaram[6], Meera R. Hameed[6], Michael F. Berger [5,6,10], David B. Solit [1,2,5], Nikolaus Schultz [7,10], Marc Ladanyi[6,10], Samuel Singer[4,12✉] & William D. Tap[1,2✉]

The genetic, biologic, and clinical heterogeneity of sarcomas poses a challenge for the identification of therapeutic targets, clinical research, and advancing patient care. Because there are > 100 sarcoma subtypes, in-depth genetic studies have focused on one or a few subtypes. Herein, we report a comparative genetic analysis of 2,138 sarcomas representing 45 pathological entities. This cohort is prospectively analyzed using targeted sequencing to characterize subtype-specific somatic alterations in targetable pathways, rates of whole genome doubling, mutational signatures, and subtype-agnostic genomic clusters. The most common alterations are in cell cycle control and *TP53*, receptor tyrosine kinases/PI3K/RAS, and epigenetic regulators. Subtype-specific associations include *TERT* amplification in intimal sarcoma and SWI/SNF alterations in uterine adenosarcoma. Tumor mutational burden, while low compared to other cancers, varies between and within subtypes. This resource will improve sarcoma models, motivate studies of subtype-specific alterations, and inform investigations of genetic factors and their correlations with treatment response.

A full list of author affiliations appears at the end of the paper.

Sarcomas are mesenchymal malignancies of the bone or soft tissue that arise in diverse anatomic locations and connective tissue types and display a range of clinical behavior from indolent to aggressive. Sarcomas are also rare tumors, representing < 1% of all malignancies in adults, though more common in the pediatric population where they represent approximately 20% of non-hematologic malignancies[1,2]. Although the diagnosis and management of sarcomas has slowly improved over the last decade, about 40% of patients with newly diagnosed sarcoma eventually die of disease. One barrier to improving outcomes in sarcoma patients is the cancer's genomic and biologic complexity, with more than 100 different subtypes now recognized by the World Health Organization[3].

Advances in clinical tumor genomic analyses have improved tumor classification; sarcomas are now classified into two broad genetic groups[4]. Sarcomas often have either simple karyotypes, harboring genetic translocations or activating mutations, or highly complex karyotypes, including numerous genomic rearrangements and large chromosomal gains and losses, commonly involving cell cycle genes such as *TP53*, *MDM2*, *RB1*, and *CDK4*. Toward identifying therapeutic targets and designing precision oncology trials based on specific sarcomas' genetic features, a comprehensive study of soft tissue sarcomas was performed by The Cancer Genome Atlas network, which analyzed 206 samples within 7 common subtypes; rarer ones were represented by as few as 5 cases[5]. Analysis of a larger cohort could define the frequency of potentially actionable alterations in rare sarcoma subtypes and broadly compare the frequency of genetic alterations across subtypes. These data, when integrated with other multiomic sarcoma studies, will facilitate better diagnostic precision, identify prognostic biomarkers, improve laboratory-based modeling of sarcomas, and generate novel hypotheses on underlying disease mechanisms.

Here, we leverage an institution-wide tumor genomic profiling initiative, MSK-IMPACT[6], to characterize genomic alterations in a large cohort of 2138 sarcomas encompassing 45 subtypes. We identify subtype-specific somatic mutations and copy number alterations, cluster tumors by genotype, and compare tumor mutation burden (TMB) and microsatellite instability among subtypes.

## Results

### Characteristics of a large and histologically diverse sarcoma cohort.
A total of 2138 bone and soft tissue sarcoma samples were analyzed together with paired normal DNA samples (Supplementary Data 1). Median patient age was 54 years (range < 1– > 90 years); 1098 (51.4%) were female. Most were primary tumors; 790 samples (36.9%) were metastases (Supplementary Table 1). The analyzed dataset included 45 distinct pathologic entities as assessed by expert sarcoma pathologists. Ninety-one of 2,138 cases in the final cohort (< 5%) were assigned an updated diagnosis after dedicated review (see Methods), often to specify the subtype within a class (e.g. alveolar rhabdomyosarcoma to replace rhabdomyosarcoma). Twenty-two subtypes were represented by ≥20 tumor samples and were therefore used as our core subtype set for analyses (Fig. 1A). Data from less represented subtypes (Fig. 1B) are included in this cohort as a resource. The most common subtypes were gastro-intestinal stromal tumor (GIST; n = 395, 18.5%), dedifferentiated liposarcoma (DDLS; n = 167, 5.4%), uterine leiomyosarcoma (ULMS; n = 165, 5.3%), and undifferentiated pleomorphic sarcoma (UPS; n = 145, 4.6%) (Fig. 1B). Rare subtypes within the core set include angiosarcoma (ANGS; n = 101, 3.2%), desmoplastic small round cell tumor (DSRCT; n = 53, 1.7%), and perivascular epithelioid cell neoplasms (PECOMA; n = 30, 0.96%) (Fig. 1B). As expected, the age distribution varied among subtypes, as did tumor

location (Fig. 1A, Supplementary Table 1). Among the more common subtypes, myxofibrosarcoma (MFS) had the oldest median age (68 years), whereas embryonal rhabdomyosarcoma (ERMS) had the youngest (8 years). Similarly, sex distribution was not uniform among subtypes (Fig. 1A); PECOMA was more common in females (23/30; 76.6%) and DSRCT more common in males (48/53; 90.5%), as was DDLS (males 115/164; 68.8%) (Fig. 1A). Among the most common subtypes, survival rate differences were most apparent starting at 3 years post-sequencing. Myxoid/round cell liposarcoma (MRLS) and GIST patients had the highest 3-year survival rates (both > 75%), whereas ANGS and alveolar rhabdomyosarcoma (ARMS) patients had the lowest (34% and 19%, respectively) (Fig. 1C).

### Identification of subtype-specific mutations.
Given the heterogeneity in sarcoma subtypes, their biologic behavior, and clinical presentation, we sought to define the genetics of individual subtypes at both the gene (mutation, gene fusion, and copy number alteration) and functional pathway levels. MSK-IMPACT identified at least one driver mutation in the majority of subtypes (Fig. 2A, Supplementary Data 1, 2, Supplementary Table 2). Overall, TMB among sarcomas was low, whereas the fraction of genome altered (FGA) in most cases was relatively high compared with other cancers, consistent with prior reports (Fig. 2A)[5]. Both varied greatly among sarcoma subtypes, especially FGA. We performed MutSig and MuSiC analyses to identify significantly recurrently mutated genes in each subtype (Fig. 2B). As expected, *TP53* and *RB1* were significantly altered across multiple subtypes, but at markedly different frequencies. Within GIST, we identified several frequently mutated genes in addition to previously known drivers such as *KIT*, *SDHA*, and *PDGFRA*[7]. These included *SETD2*, which encodes a histone methyltransferase (4%), *MAX*, which encodes a MYC binding partner and transcription factor (4%), and *MGA* (3%), whose product binds the MAX-MYC complex[8].

Additional subtypes with recurrently mutated genes of potential biologic or clinical relevance included ANGS (n = 101), in which we identified recurrent mutations in receptor tyrosine kinases involved in angiogenesis including *KDR* (VEGFR2; 19%) and *FLT4* (VEGFR3; 9%) as well as another receptor tyrosine kinase, *EPHA5* (9% of cases). The mutations in *EPHA5* and *FLT4* were all variants of unknown significance (VUS). The VUS in *FLT4* all affect the kinase domain or the C-terminus, implying a possible functional consequence (Fig. 2C). In Ewing sarcoma (ES; n = 99), 10% of samples carried mutations in *STAG2*, which encodes a cohesion complex component, confirming prior reports[9]. In ULMS, *MED12*, which encodes a member of the transcription elongation complex, was altered in 16% of cases, most frequently missense mutations at glycine 44, as reported previously (Fig. 2B)[10].

In PECOMA, SFT, LMS, ULMS, and ES, driver mutations were represented in a cancer cell fraction (CCF) of close to 1.0, suggesting that these represent a large clonal population (Fig. 2D). The CCF for VUS was overall similar to that of drivers within most subtypes, with the exception of MFS, OS, and UPS, which suggests that in some cases these VUS could have an unrecognized function, calling for further studies to determine their roles in oncogenesis and progression.

### Chromosomal gains or losses are shared across multiple subtypes and whole-genome doubling is associated with outcomes.
As many sarcomas are driven by copy number alterations, we analyzed these changes across the whole cohort, including in subtypes not classically thought to be driven by them (Fig. 3). For instance, in GIST patients (evaluable n = 371), there were

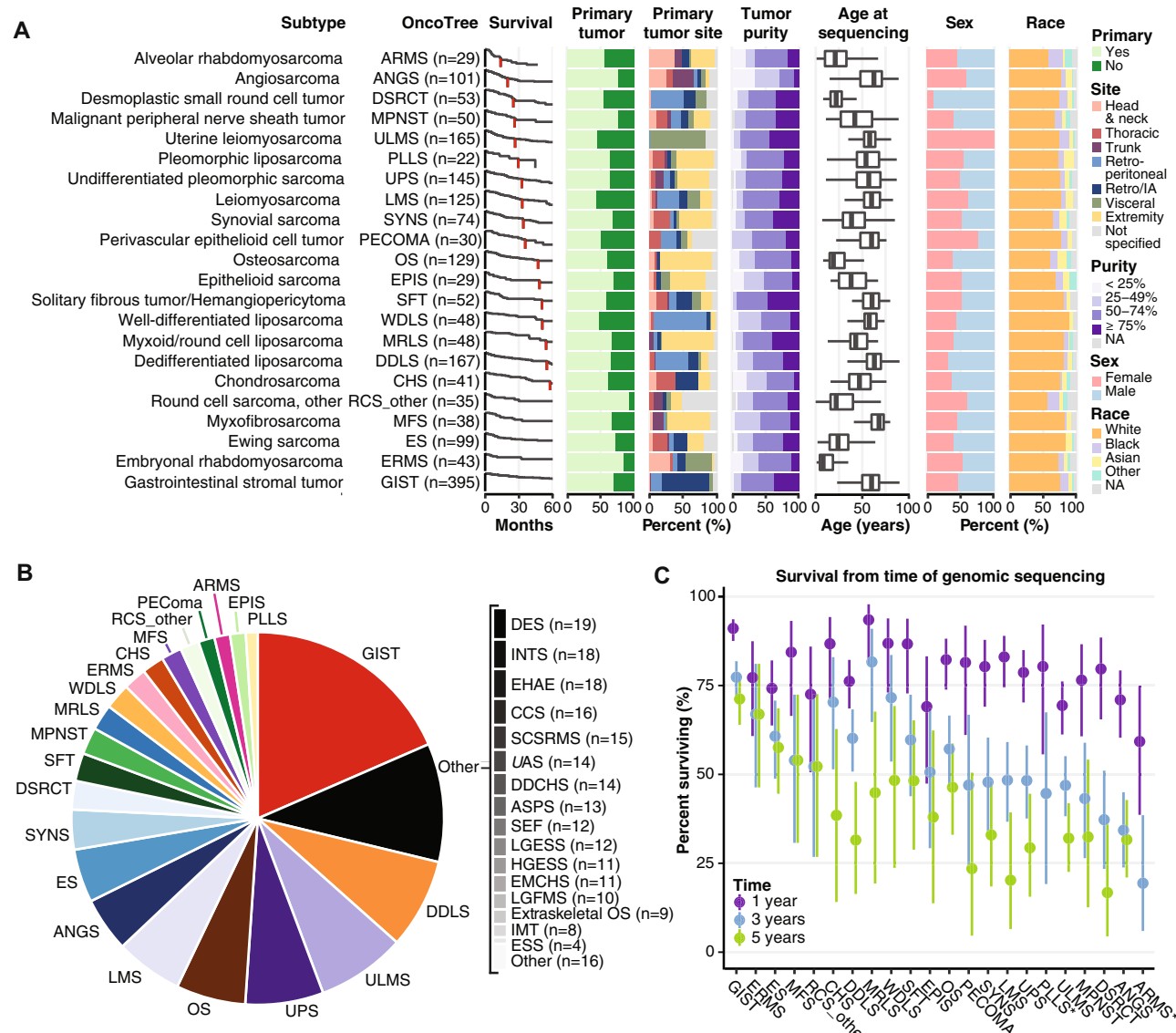

**Fig. 1 Analysis of 2138 sarcoma samples reveals variation in patient characteristics among subtypes.** This analysis includes 2138 bone and soft tissue sarcoma samples, each from distinct patients. Subtypes with ≥20 samples in the dataset are displayed. **A** Distribution of number of samples, survival from time of sequencing, sample type (primary or metastatic site), tumor site, sample purity, age, sex, and self-reported race in each subtype. Retro/IA, retroperitoneal or intrabdominal. NA, not applicable. In the age plot, box boundaries indicate 25th and 75th percentiles, interior lines medians, and whiskers 1.5 times the interquartile range. **B** Overall distribution of sample number for the entire cohort. **C** 1, 3, and 5-year survival from the time of sequencing. *, 5-year survival = 0. Vertical lines indicate 95% confidence intervals. DES desmoid tumor, ESS endometrial stromal sarcoma, INT intimal sarcoma, LGFMS low-grade fibromyxoid sarcoma, EMCHS extraskeletal myxoid chondrosarcoma, HGESS high-grade endometrial stromal sarcoma, LGESS low-grade endometrial stromal sarcoma, SEF sclerosing epithelioid fibrosarcoma, ASPS alveolar soft part sarcoma, DDCHS dedifferentiated chondrosarcoma, UAS uterine adenosarcoma, SCSRMS spindle cell/sclerosing rhabdomyosarcoma, CCS clear cell sarcoma, EHAE epithelioid hemangioendothelioma.

frequent copy number loss events involving chromosomes 1, 14, 15, and 22 (Fig. 3A). Translocation-driven sarcomas, e.g. ES, DSRCT, and SYNS, exhibited highly recurrent copy number changes, indicating that there may be additional relevant genetic events beyond the driver translocations (Fig. 3B). We identified a diversity of chromosome arms (e.g. 5p, 8q, and 10p) that were recurrently affected by copy number variation across multiple common subtypes (Fig. 3B). Of note, 12q amplifications in DDLS and WDLS patients were not wide enough to be called in arm-level analysis (Fig. 3B, left), though they were clearly observed as a strong focal event in copy number profiles (Fig. 3A, B, right). In most cases, these arm-level copy number events were not linked to a specific gene. However, there were some exceptions including significant gains of *MYC* on chromosome 8q24 in OS, EPIS,

ERMS, and ANGS, as well as significant gain of a gene encoding a negative regulator of NF-kB signaling, *TNFAIP3*, in DDLS (Fig. 3B). As expected, we observed more widespread copy number changes in classically copy number-driven subtypes such as LMS, ULMS, MFS, and OS compared with the rest of the cohort.

Despite sharing *CDK4* and *MDM2* amplification events, DDLS is more aggressive than WDLS and has increased risk for distant spread[11]. Therefore, we compared rates of amplifications between WDLS and DDLS, and found greater rates of amplification of the oncogenes *GLI1* (8.5% vs. 25.3%), *TERT* (6.3% vs. 14.4%), and *JUN* (0% vs. 13.8%) in DDLS. The Jun transcription factor positively regulates the expression of cyclin D1, a CDK4/6 cyclin partner, and amplification of *JUN* in DDLS is associated with a

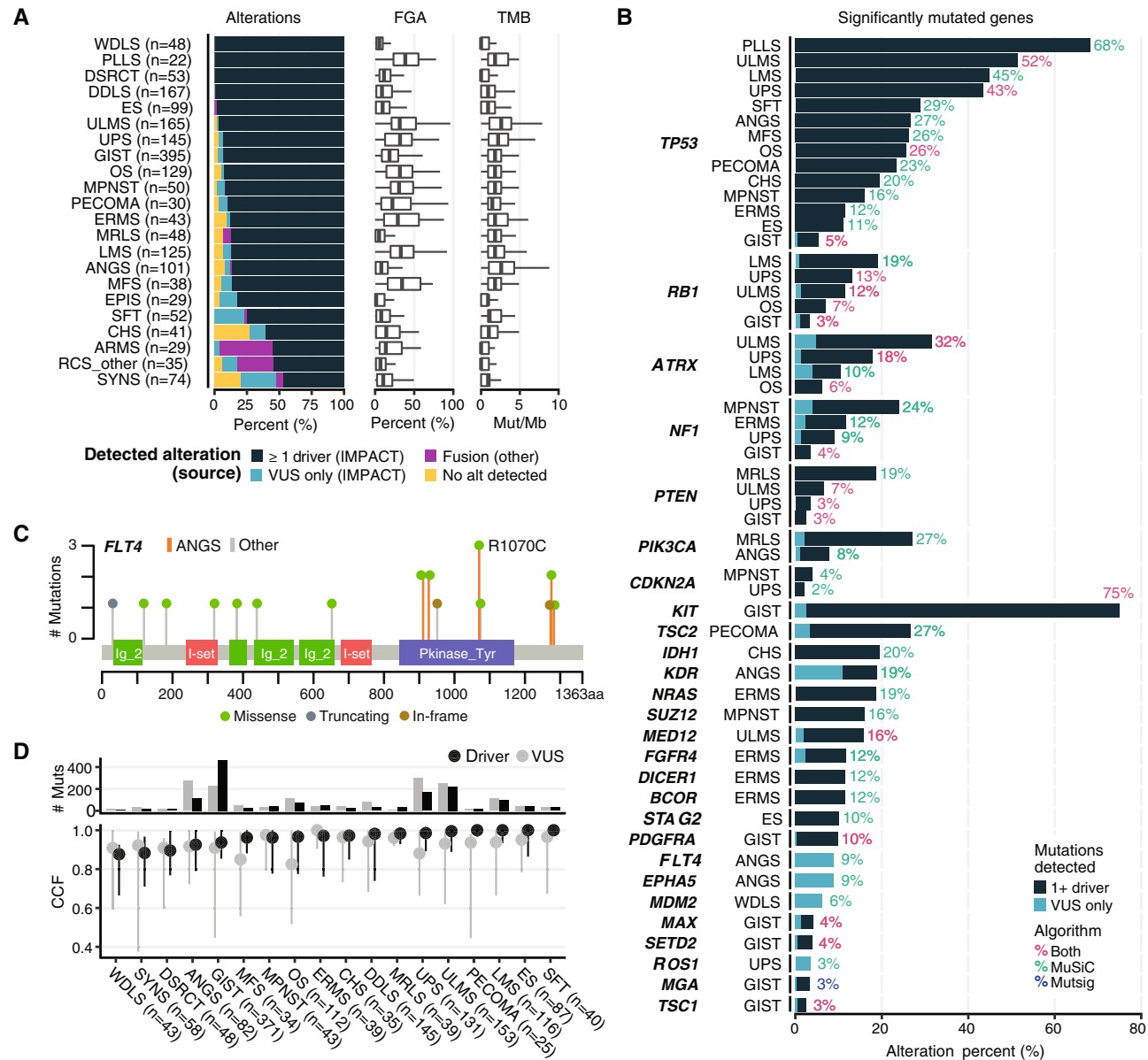

**Fig. 2 Mutation analysis by subtype. A** Alteration type and frequency, fraction of genome altered (FGA) and tumor mutation burden (TMB) by subtype. Oncogenic fusions detected by MSK-IMPACT are classified as drivers. In the FGA and TMB plots, box boundaries indicate 25th and 75th percentiles, interior lines medians, and whiskers 1.5 times the interquartile range. VUS variant of unknown significance. **B** Significant mutations were identified in all subtypes with $n \geq 20$ in our dataset using both MutSig and MuSiC analysis. Percentages indicate the percentage of samples with an oncogenic mutation in the corresponding gene. **C** *FLT4* mutation type, frequency, and location in ANGS vs. other subtypes. **D** Cancer cell fraction (CCF) and number of mutations for driver mutations and VUS by subtype. Circles indicate medians and vertical lines interquartile ranges.

more aggressive phenotype[5], calling for investigation of whether *CDK4* and *JUN* co-amplification drives progression to DDLS or modulates response to *CDK4* inhibition. Amplification of the GLI1 transcription factor, downstream of Sonic Hedgehog (Shh) signaling, has previously been reported[12]; this confirmation provides further rationale for studying Shh pathway inhibition in DDLS. *GLI1* amplification and *JUN* amplification were mutually exclusive.

We also analyzed whole genome doubling (WGD) events across subtypes and compared them to data from a pan-cancer analysis where WGD was associated with decreased overall survival[13]. In that study, WGD was further associated with defects in cell cycle regulation and increased proliferative rates, which could explain differences in patient outcomes. Therefore, WGD warrants further study as a potential prognostic biomarker

in sarcomas on a subtype-specific basis. OS, UPS, ERMS, and MPNST had high frequencies of WGD, all around 50%, ranking among the highest even among a wide variety of cancers for which WGD was previously analyzed (Fig. 3C; Supplementary Fig. S1A)[13]. In keeping with the notion that MFS is on a genetic continuum with UPS[5], UPS and MFS had similar WGD frequencies. Despite being copy number alteration (CNA)-driven, WDLS and DDLS had lower rates of WGD frequency, as did many translocation-driven subtypes including SYNS, ES, DSRCT, and MRLS. In sarcomas, WGD was associated with worse overall survival among metastatic ($p = 0.042$) but not primary cases ($p = 0.391$; Supplementary Fig. S1B). Among specific subtypes, WGD was associated with worse overall survival (from time of sequencing) in metastatic UPS ($p = 0.022$; Fig. 3D), but not MFS ($p = 0.78$; Supplementary Fig. S1C).

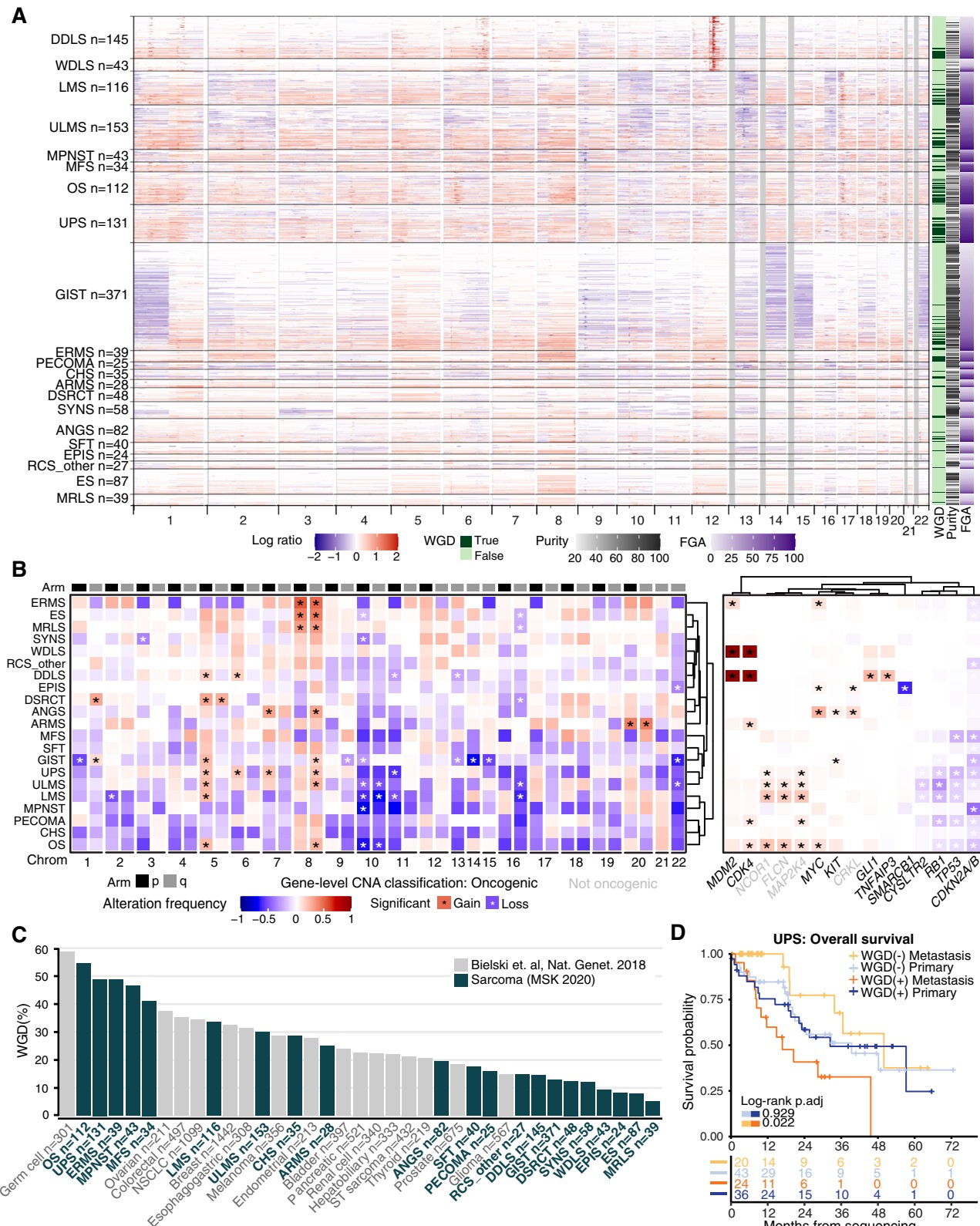

**Fig. 3 Copy number changes by subtype.** Copy number alteration (CNA) and whole-genome doubling events (WGD) compared across subtypes.
**A** Individual sample CNA across the genome for each subtype. WGD, fraction genome altered (FGA), and purity are shown at right. **B** Aggregate arm-level (left) and gene-level events (right) grouped by subtype. *significant change based on Bonferroni corrected *p*-values. Significance was evaluated by random permutations testing. Oncogenic (bold) vs. non-oncogenic CNA classifications according to OncoKB. **C** Frequency of WGD by subtype (green) compared to other cancers with ≥200 samples available for comparison (gray)[13]. NSCLC, non-small cell lung cancer; ST, soft tissue. **D**, Overall survival based on WGD status within primary and metastatic samples. p.adj, adjusted *p*-value.

**Alterations in epigenetic, cell cycle, and *PI3K* pathways are common**. Pathway-specific analyses within each sarcoma subtype for which ≥20 samples were available (Fig. 4A, genes in each pathway listed in Supplementary Data 3) revealed that a number of pathways important in carcinomas were infrequently altered in sarcoma, including TGFβ, WNT, Hippo, Notch, and *NRF2* (Fig. 4A, right panel). By contrast, the cell cycle and *TP53* pathways were altered in at least half of samples in 8 of the 22 most common subtypes. For instance, DDLS and WDLS demonstrated cell cycle or *TP53* pathway alterations in 214/215 (99%) of samples as expected, most commonly through co-amplification of *CDK4* and the gene encoding the E3 ubiquitin ligase that targets p53 for degradation, *MDM2* (Fig. 4A)[5,14–17]. Many of the sarcomas with infrequent alterations (<10%) in the cell cycle and *TP53* pathways were driven by translocations (e.g. MRLS, DSRCT, SYNS) or alteration in genes encoding components of the SWI/SNF remodeling complex (epithelioid sarcoma [EPIS]) (Fig. 4A), highlighting a distinct mechanism of pathogenesis. An exception was solitary fibrous tumor (SFT), which is driven by the *NAB2-STAT6* fusion oncogene, and has oncogenic *TP53* alteration in 28% of cases (Fig. 4A)[18].

The PI3K pathway was frequently altered in some of the most prevalent subtypes in our dataset including MRLS (41%), PECOMA (40%), ULMS (30%), pleomorphic liposarcoma (PLLS; 22%), UPS (20%), and soft tissue leiomyosarcoma (LMS; 20%) (Fig. 4A, right panel). Among these subtypes, *PTEN* and *PIK3CA* were the most frequently affected genes except in PECOMA where *TSC2* loss of function alterations were most common (30%) (Fig. 4A, B). *PTEN* loss of function alterations predominated in LMS and ULMS (14% and 21%, respectively), whereas in MRLS, *PIK3CA* mutations were most frequent, occurring in 25% of cases, consistent with our prior findings[19]. In MRLS, *PTEN* loss was observed in 21% of cases, some of which were concurrent with *PIK3CA* mutations (4 *PIK3CA* mutations in 10 *PTEN* loss cases). In contrast, in UPS, *PTEN* alterations were identified in 8% of samples and *PIK3CA* in 3%; only 1 of the 11 cases with a *PTEN* alteration had a concurrent *PIKC3A* mutation. Notably, *PTEN* loss of function has also been proposed as a predictor of non-response to immune checkpoint inhibition in ULMS[20]. In ANGS, in which the PI3K pathway, specifically *PIK3CA*, is known to be altered[21], we identified oncogenic *PI3KCA* alterations in 6% of cases. This is lower than the previously reported rate of 21%, perhaps owing to a larger sample size in our study ($n = 101$ vs. 47) and our exclusion of VUS.

Because a pan-cancer MSK-IMPACT analysis identified *TERT* promoter mutations in a subset of sarcomas[22], we investigated *TERT* alteration frequency as a function of sarcoma subtype (Fig. 4C). We identified oncogenic *TERT* amplifications in 44% (8/18) of intimal sarcoma (INTS) and *TERT* promoter mutations in 79% (38/48) of MRLS, 46% (24/52) of SFT, and 35% (5/14) of dedifferentiated chondrosarcoma (DDCHS). In DDLS, oncogenic *TERT* promoter alterations were present in 16% of samples (27/167) and were almost entirely amplifications ($n = 24$). *TERT* copy number alterations have not yet been described in INTS, perhaps due to the low incidence of this rare subtype. The *TERT* locus is distinct from that of the *MDM2* and *CDK4* amplifications[23] that are hallmarks of INTS, implicating *TERT* amplification as a potential independent contributor to pathogenesis.

Alterations in DNA damage repair (DDR) pathway genes have been associated with the development of sarcomas[24], and are of particular clinical interest as PARP inhibition has activity in select carcinomas with homologous recombination deficiency and immune checkpoint blockade has activity in certain tumors with microsatellite instability[25,26]. Our analysis of DDR pathway alterations found that 9.6% of all samples harbored an oncogenic somatic alteration in the DDR pathway. Among subtypes with more than 20 samples, the frequency of DDR gene alterations was highest in ULMS (24%), MPNST (16%), PLLS (13%), PECOMA (13%), ANGS (13%), LMS (10%), and OS (10%) (Fig. 4A, right panel). The most frequently altered genes across subtypes were *BRCA2* (1.4% of all samples), *RAD51B* (1.1%), *CHEK2* (1.0%) *ATM* (0.9%), *FANCA* (0.6%), and *RAD51* (0.6%). Consistent with a previous report in uterine sarcomas[27], nearly half of *BRCA2* (41%) and *RAD51B* (47%) alterations occurred in sarcomas of uterine origin, with *RAD51B* or *BRCA2* each mutated in 7% of ULMS cases. Similarly, 35% of the 14 uterine adenosarcomas also had an altered DDR gene, all deep deletions. Five percent of ANGS had oncogenic mutations and another 5% had a VUS in *ATM*. Given the known association of a subset of ANGS with prior ionizing radiation, which suggests a role for DNA damage in the pathogenesis of ANGS, mutations in *ATM*, which is important for DNA damage repair, may represent a convergent or synergistic mechanism for the accumulation of DNA damage in ANGS. Of the 15 sarcomas (0.7%) with an altered mismatch repair (MMR) gene (*MLH1, MSH2, MSH6,* or *PMS2*), one (LMS) was microsatellite instability (MSI)-high by MSIsensor and had a high TMB.

Epigenetic dysregulation contributes to the pathogenesis of several sarcoma subtypes[28]. In SYNS, EPIS, malignant rhabdoid tumors, and MPNST, this occurs through alterations of chromatin-remodeling and -modifying complexes; in chondroblastoma, giant cell tumors of bone, and potentially in some CHS, UPS, and osteosarcoma through oncogenic histone mutations. In light of emerging pharmacologic strategies to study and therapeutically target epigenetic regulatory proteins, we identified sarcoma subtypes harboring by epigenetic pathway alterations (Fig. 4A; Supplementary Data 3)[29]. As expected, 75% of EPIS had loss-of-function deletions, truncating mutations, or intragenic fusions in *SMARCB1*. In addition, the epigenetic pathway was one of the most altered pathways among the highly prevalent subtypes in our dataset. Pathogenic alterations in epigenetic pathway genes (Supplementary Data 3) were observed in 64% of MPNST, 49% of ULMS, 45% of PLLS, 43% of CHS, 42% of UPS, 36% of MFS, and 32% of OS (Fig. 4A). By contrast, these alterations were infrequently observed (< 10%) in WDLS, ARMS, and MRLS, suggesting subtype specificity.

We determined the association with specific subtypes of epigenetic pathway genes contributing to a specific biochemical function (e.g. DNA methylation, chromatin remodeling) and complex (e.g. PRC1, PRC2, MLL3/4) (Fig. 4D; Supplementary Fig. 2; Supplementary Table 3). Genes involved in histone modification were altered in 48% of MPNST, 42% of sclerosing epithelioid fibrosarcoma (SEF), 36% of uterine adenosarcoma (UAS), and 36% of high-grade endometrial stromal sarcoma (HGESS). ERMS had frequent alterations in *BCOR*, which encodes a transcriptional co-repressor and non-canonical PRC1 complex member (19% total, 16% oncogenic), which were mutually exclusive with *DICER1* alterations (12% oncogenic). Both alterations were more prevalent in our population than in prior studies[30,31].

Genes involved in chromatin remodeling were altered at similarly high frequencies: 76% of EPIS, 39% of ULMS, 26% of UPS, 24% of MFS, and 18% of MPNST. In a significant portion of these cases, alterations in the histone chaperone-encoding gene *ATRX* drove these high rates (Supplementary Fig. 2). We also note the unexpected finding that UAS ($n = 14$), a rare sarcoma subtype, had oncogenic alterations in genes encoding subunits of the SWI/SNF chromatin remodeling complex in 43% of patients, with *ARID1A* and *PBRM1* most frequently affected (Fig. 4D, Supplementary Fig. 2). Interestingly, UAS also had alterations in histone-modifying genes in 36% of cases.

As epigenetic alterations are more frequent in DDLS (25%) than WDLS (8%) and we have previously found epigenetic

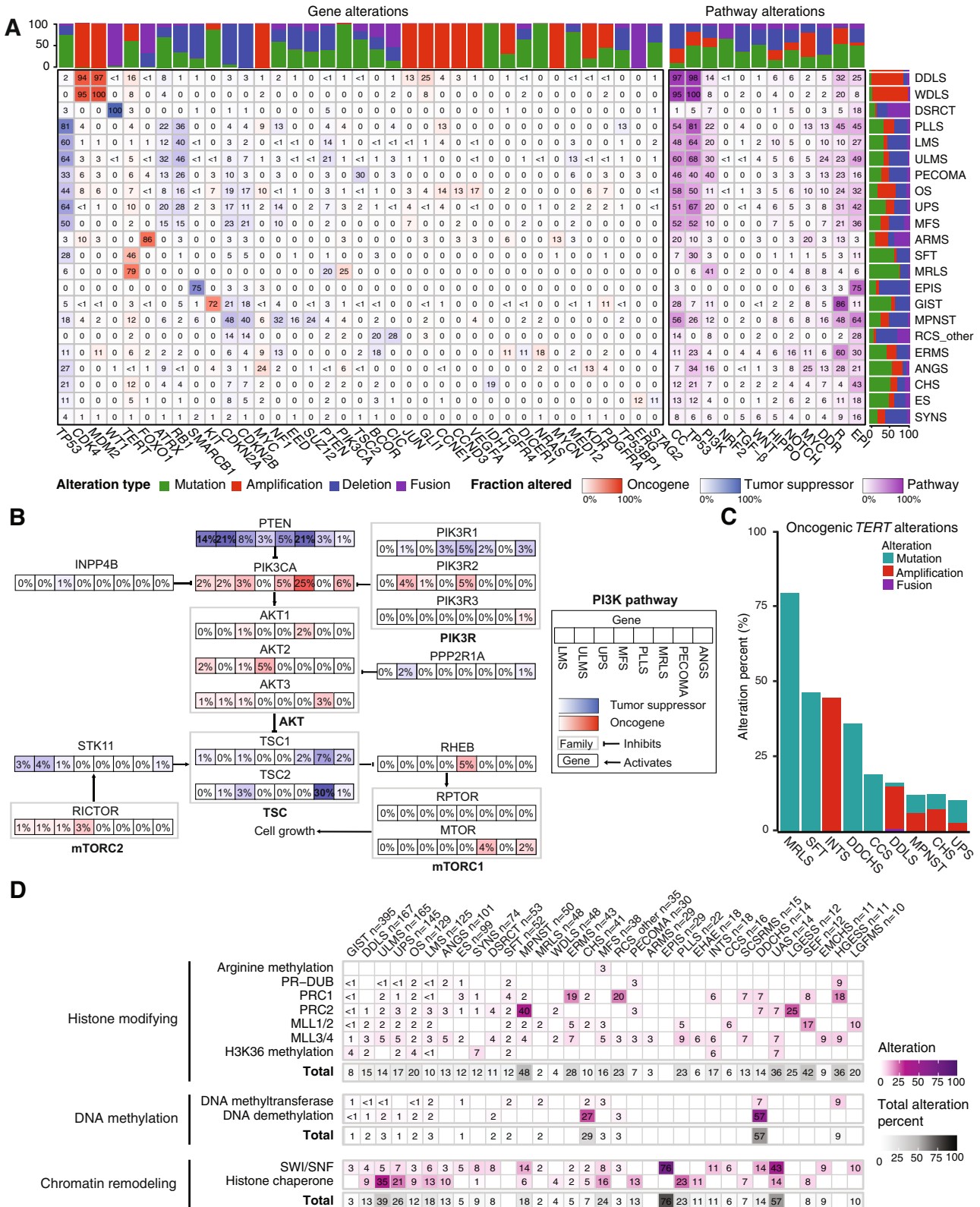

**Fig. 4 Integrated pathway analysis. A** Oncogenic alterations within each of 12 pathways with relevance to cancer biology in each subtype. Numbers in each cell indicate percentage of samples harboring alterations. Stacked bar graphs indicate the distribution of the type of oncogenic alteration per gene or pathway (top) or subtype (right). CC, cell cycle; DDR, DNA damage repair; EPI, epigenetic. **B** PI3K pathway alterations in specific subtypes. The percentage of samples with an alteration in a specific gene in each subtype is indicated in each box. **C** Oncogenic *TERT* alterations in each of the 9 most altered subtypes. **D** Oncogenic epigenetic pathway alterations by subtype, grouped by complex and/or biochemical function of the encoded protein. Totals include all alterations in genes that belong to a parent category, not only those affecting specific complexes listed.

dysregulation to contribute to DDLS[32], we further examined differences between DDLS and WDLS. Histone-modifying and histone chaperone/chromatin-remodeling alterations occur in 15% and 13% of DDLS cases, respectively, compared with 4% each of WDLS (Fig. 4D). This suggests that loss of epigenetic regulation could be an important contributor to dedifferentiation.

We also examined epigenetic pathways without filtering for alterations already established as oncogenic, which is a strategy we recently employed to generate hypotheses in an analysis of genetic alterations in OS (Supplementary Fig. 3)[33]. This analysis identified the histone methyltransferase-encoding gene *KMT2D/MLL4* as more frequently altered in MFS (16%) compared with other subtypes including the closely related UPS (6%)[5]. Histone-modifying enzyme genes were altered in 20% of SYNS, among which *KMT2B* and *SETD2* were altered in 6% and 7% of samples, respectively. We also found that *NCOR1*, the protein product of which complexes with *HDAC3* and other deacetylases to co-repress the activity of transcription factors such as the retinoic acid receptor and thyroid hormone receptor[34], was altered in 10% of ULMS, 19% of LMS, and 21% of OS, mostly through amplification. NCOR1 has also been shown to regulate transcription factors important in mesenchymal lineages including the MEF2 family and PPARγ, which regulate myo- and adipogenesis, respectively[35]. Several other genes within the same cytoband as *NCOR1*, 17p12-p11.2, were co-amplified, including *FLCN*, *MAP2K4*, *AURKB*, and *ALOX12B* (Supplementary Fig. 4A). Amplifications of *MYOCD*, whose genomic location is within a region previously found to be amplified in LMS[36], were not detected because this gene is not represented on the MSK-IMPACT panel. Except for *ALOX12B*, the 17p copy number gains of MSK-IMPACT-assessed genes were associated with increased gene expression in the sarcoma TCGA analysis (Supplementary Fig. 4B). Thus, one or more of these genes could play a pathogenic role.

**Gene and pathway alterations co-occur in subtype-specific contexts**. To better understand how gene- and pathway-level alterations interact, we analyzed their co-occurrence and mutual exclusivity (Fig. 5A). As expected, *KIT* and *PDGFRA* alterations were mutually exclusive in GIST and *CDK4* and *MDM2* co-occurred in DDLS. In OS, *KDR* alterations co-occurred with *KIT* and *PDGFRA*, as did the latter two with each other, suggesting dysregulation of signaling through these 3 RTK genes located at the 4q12 locus[33]. *TP53* alterations were mutually exclusive with *CDKN2A/B* in GIST and ULMS, but not in UPS. In ES and SFT, *TP53* alterations co-occurred with *STAG2* and *TERT* alterations, respectively, suggesting context dependence for alterations co-occurring with *TP53*. In UPS, *ATRX* and *NF1* alterations, which are mostly loss-of-function events, were mutually exclusive, suggesting biologically different subgroups.

At the pathway level, cell cycle and DDR pathway alterations significantly co-occurred (false discovery rate [FDR] < 0.05) with those in other pathways. For instance, cell cycle alterations co-occurred with *MYC* pathway alterations in GIST, with PI3K pathway alterations in GIST and ULMS, and with RTK/RAS alterations in GIST, OS, and SYNS (Fig. 5A). DDR pathway alterations co-occurred with *MYC* pathway alterations in ULMS and MRLS, epigenetic pathway alterations in GIST and DDLS, Hippo pathway alterations in DDLS, and cell cycle alterations in ES. There were no examples of significant mutual exclusivity at the pathway level.

**ATRX is recurrently altered in multiple subtypes**. *ATRX* stood out across subtypes as frequently affected by loss-of-function events (Fig. 5B); this gene was altered in ≥ 10% of cases in 7 subtypes: ULMS, PLLS, UPS, MFS, PECOMA, LMS, and ANGS. In ULMS, which had the highest rate of *ATRX* alterations, the

frequency was roughly 1 in 3 cases. That *ATRX* loss-of-function events occur in both copy number- and translocation-driven subtypes, although at lower frequency in the latter, raises the possibility that they may serve a fundamental role in the biology of a molecular subset of these subtypes. *ATRX* loss-of-function mutations were more frequent than deletion events, independent of subtype, despite the overall low mutation rate. Our analysis also captured intra- and intergenic *ATRX* fusion events.

**Unsupervised clustering of subtypes reveals genomic groupings distinct from histologic identities**. To assess genetic similarities among subtypes, we grouped samples on the basis of genetic alterations by unsupervised clustering (Fig. 5C), which generated 17 distinct clusters subsequently named according to their pre-vailing subtype and/or genetic feature. Some subtypes and clusters were closely associated (Fig. 5D). These included EPIS and the *SMARCB1* cluster, DSRCT and *WT1*, WDLS, DDLS, PAOS, and INTS with *MDM2-CDK4*, and MRLS and SFT with *TERT*. These groupings largely reflect known or presumed drivers in these subtypes, reinforces their central roles therein, and supports the rationale for this clustering approach.

However, many subtypes were markedly varied in their cluster association. To quantify this heterogeneity, for each subtype we also assigned an entropy score with respect to the clustering assignments (Fig. 5D). WDLS, DDLS, and DSRCT had the lowest entropy, suggesting relatively uniform genomic profiles within each subtype, whereas ULMS, UPS, and OS had high entropy, suggesting that these pathologically defined entities harbor multiple distinct genetic variants.

One cluster lacked any predominantly altered gene. This 'other' cluster included the majority of samples in many histotypes (e.g. ERMS and UAS) but also included samples from multiple subtypes that are represented more commonly in other clusters (e.g. LMS and UPS), suggesting that there may be genetic outliers among those subtypes. Interestingly, while roughly 75% of CHS samples are in the 'other' cluster, in contrast DDCHS, which is thought to arise from CHS and is clinically and morphologically similar to OS, has few samples in the 'other' cluster and instead falls into clusters shared with OS suggesting a possible genetic shift underlying the dedifferentiated phenotype. In addition, while there were some similarities in the clustering profiles of OS and extraskeletal OS, which arises outside of bone and is treated as a soft tissue sarcoma, extraskeletal OS has a substantially greater fraction of samples in the *TP53-ATRX-RB1* cluster than OS and all other subtypes suggesting a distinct genetic subgroup.

**Tumor mutational burden is low but heterogeneous between and within subtypes**. Two recent immune checkpoint blockade trials in sarcoma demonstrated low overall response rates, though rates varied among subtypes[37,38]. Thus, predictive biomarkers for response to checkpoint blockade are needed to deconvolute this heterogeneity and aid in the design of future clinical trials. As microsatellite instability predicts response to pembrolizumab[25], and tumors with high TMB are more likely to respond to immune checkpoint blockade[39], we determined MSI status and TMB for each subtype (Supplementary Fig. 5A, B, Supplementary Data 1).

While the median TMB for sarcomas was low compared to many carcinomas[5], there was considerable heterogeneity within and between the more common subtypes in our cohort (inter-subtype median range 0.9–3.0) (Supplementary Fig. 5A). The median TMB was greatest in ANGS (3.0), UPS (2.6), and ULMS (2.6) and lowest in WDLS (0.9), EPIS (0.9), and RCS (other) (0.9). However, in certain subtypes, the distribution of TMB had a long upper tail and was skewed towards higher TMB (Supplementary Fig. 5A). TMB was ≥ 5 mut/Mb in 25% of ANGS, 15% of ULMS

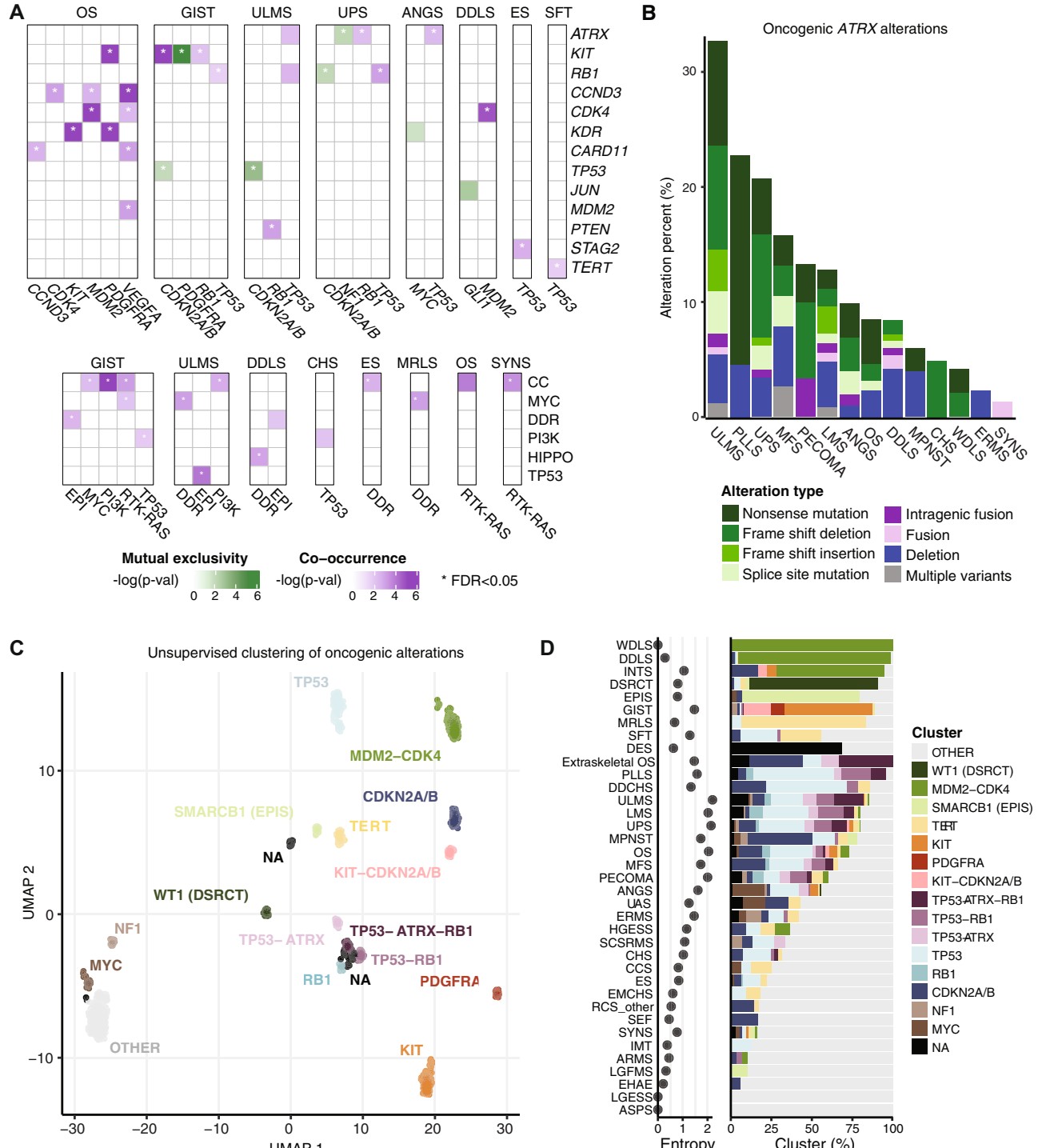

**Fig. 5 Mutual exclusivity, co-occurrence, *ATRX* alterations, and unsupervised clustering based on genetic signatures. A** Co-occurrence and mutual exclusivity of gene- (top) and pathway-level (bottom) alterations in each subtype with significant findings shown. Significance was evaluated by two-sided Fisher's exact test. EPI, epigenetic; DDR, DNA damage repair. **B** Frequency and types of oncogenic *ATRX* alterations in each of the 14 most altered subtypes. **C** Unsupervised clustering of all samples based on oncogenic alteration patterns. **D** Subtype-specific cluster associations and entropy scores. For clarity, subtypes with *n* > 5 are displayed.

and UPS, and 13% of ERMS. Only two subtypes had ≥5% of samples with a TMB of ≥10 mut/Mb: ANGS (7.6%) and UPS (6.7%).

Only 5 of 1893 samples evaluable for MSI status were MSI-high (by MSIsensor score ≥10), including one UPS, one LMS, and 3 ULMS (Supplementary Fig. 5B). Of these, 4 were confirmed to be MSI-high by a conventional PCR-based MSI assay. MSIsensor

scores varied widely between subtypes (Supplementary Data 1). Overall, while microsatellite instability corresponded with high TMB, the inverse was not true.

To understand mechanisms contributing to extensive genetic alterations, we examined mutational signatures in samples with ≥15 single nucleotide variants (SNVs) (Supplementary Fig. 5C). A UV mutational signature was observed in a subset of ANGS and

was most strongly observed in samples at the highest end of the TMB spectrum within that group. Of these 16 samples, 11 had a head and neck primary site. Interestingly, a subset of UPS also harbored a UV signature and a higher TMB, while another subset of highly mutated UPS was dominated by an aging signature, suggesting alternative mechanisms for high TMB within UPS.

**Targeted sequencing reveals actionable alterations**. Toward improved detection of targetable alterations for each subtype and patient, we analyzed genetic alterations by actionability according to OncoKB (Fig. 6A–D)[40]. As expected, Level 1 alterations, defined as FDA-recognized biomarkers for response to an FDA-approved drug, were most frequent in GIST due to *KIT* and *PDGFRA* mutations (Fig. 6A). Similarly, Level 1 *SMARCB1* deletion was noted in 66% of EPIS. Level 2 alterations, defined as guideline-supported standard-of-care biomarkers for an FDA-approved drug, were seen in > 90% of WDLS and DDLS related to *CDK4* amplification. In the same subtypes, *MDM2* amplifications in >90% of cases were deemed Level 3A, for which compelling evidence supports use as a predictive biomarker for an existing drug. Many other observed alterations were classified as Level 3B, defined as standard-of-care or investigational biomarkers that predict response to an FDA-approved or investigational drug in another cancer. Notable examples included a combined 37% prevalence of actionable *TSC1/2* deletions in PECOMA, *IDH1/2* alterations in 27% of CHS, and targetable PI3K pathway (*PIK3CA*, *ATK1*, *MTOR*, or *TSC1*) alterations in a collective 31% of MRLS (Fig. 6B). Notably, 21% of MRLS cases had Level 4 *PTEN* deletions, for which compelling biological evidence supports their use as a predictive biomarker. Other intriguing Level 4 alterations included somatic *NF1* deletions in MPNST (32%), UPS (14%), ERMS (14%), and PLLS (14%), and *CDKN2A* deletions in many subtypes at a rate of up to 48% as seen in MPNST.

## Discussion

To better understand genetic heterogeneity in sarcomas, we analyzed prospectively generated tumor next generation sequencing data from a cohort of 2138 sarcoma samples representing 45 histological subtypes. Across all subtypes, the most common alterations we identified were in cell cycle control and *TP53*, receptor tyrosine kinases/PI3K/RAS, and epigenetic regulators. Subtype-specific associations included *TERT* amplification in intimal sarcoma and SWI/SNF complex alterations in uterine adenosarcoma. Tumor mutation burden varied widely between and within subtypes.

Epigenetic pathway mutations frequently occurred in many subtypes in our cohort, in keeping with an emerging recognition of epigenetic dysregulation as an important factor in the pathogenesis of sarcomas[28]. A common epigenetic pathway alteration was amplification of *NCOR1*, particularly in ULMS, LMS, and OS. *NCOR1* encodes a transcriptional corepressor that regulates transcription factors specific to mesenchymal lineages and can suppress differentiation when overexpressed[35,41]. If amplification of *NCOR1* correlates with increased protein levels in these sarcomas, which RNA sequencing data suggests it may, this could lead to altered differentiation and transcriptional programs. Moreover, since the activity of NCOR1 is modulated by PI3K/Akt-mediated control of nuclear localization, both inhibition of that pathway and of HDAC3 warrant further exploration as potential therapeutic strategies in *NCOR1*-amplified ULMS, LMS, or OS[42].

In uterine adenosarcoma, we identified genetic alterations in the SWI/SNF chromatin remodeling complex in 43% of cases, mostly loss-of-function alterations in *ARID1A* or *PBRM1*. Uterine

adenosarcoma is a rare subtype composed of both sarcomatous stroma and benign epithelium, which can behave aggressively, especially in the setting of sarcomatous overgrowth[43]. Given the role of epigenetic regulation in determining differentiation, impaired SWI/SNF function could contribute to this phenotype. Histone mutations have been observed in ovarian carcinosarcoma, suggesting that epigenetic dysregulation may be a common mechanism for impaired lineage commitment in Müllerian tumors[44]. Given synthetic lethality between the loss of the SWI/SNF component-encoding gene *SMARCB1* in epithelioid sarcoma and EZH2 inhibition with the now FDA-approved drug tazemetostat, EZH2 inhibition may represent a future therapeutic strategy in uterine adenosarcoma[45].

We also analyzed genes involved in maintenance of telomeres, whose tumor-suppressive function is dependent on the silencing of *TERT*, which encodes a reverse transcriptase and core component of telomerase. Mutations in the *TERT* promoter, first identified in melanoma, lead to increased transcription of the *TERT* gene[46,47]. Within our cohort, *TERT* amplification occurs in 44% of intimal sarcomas. Whether this amplification leads to increased expression of the TERT gene product should be investigated, as *TERT* overexpression is known to be oncogenic in certain contexts[48]. In addition, our data validate prior findings of *TERT* mutations in MLPS and SFT. However, the rate in SFT was greater than observed in prior studies[49], which may be explained by differences in disease aggressiveness, as *TERT* mutations associate with worse prognosis[50].

While we included *ATRX* in the epigenetic pathway gene list owing to its product's fundamental role, along with *DAXX*, as a histone variant H3.3 chaperone, ATRX also participates in other pathways including telomerase-independent alternative lengthening of telomeres (ALT), which has been observed in a number of soft tissue sarcomas including UPS and liposarcoma[51]. Because UPS and liposarcoma also harbor *TERT* alterations in a largely non-overlapping pattern, these sarcomas may acquire the ability to aberrantly maintain telomeres through multiple independent mechanisms. In addition to epigenetic and ALT functions, ATRX helps maintain genomic integrity[52]. Because of the diversity of the physiologic functions of ATRX, the role(s) of *ATRX* alterations in sarcomagenesis are difficult to predict a priori. Thus, developing tools such as patient-derived cell lines and xenografts to study the impact of these alterations on ATRX-dependent functions will be informative. Given the relative frequency of *ATRX* alterations and the inclusion of *ATRX* on MSK-IMPACT and other targeted sequencing platforms, such investigations are eminently feasible.

Toward identifying predictive biomarkers for response to immune checkpoint blockade in sarcoma, we analyzed the distribution of MSI-H and high TMB, which are associated with response to these agents in other solid tumors. Almost none of the samples had microsatellite instability and there were relatively few samples with high TMB. However, the upper tail of TMB was relatively long in certain subtypes such as UPS, ANGS, and ULMS. Moreover, we do not yet know whether the TMB cutoff of 10 mutations per megabase, which defines high TMB for carcinomas and predicts response to immune checkpoint blockade, is the appropriate threshold for TMB as a predictive biomarker in mesenchymal neoplasms, let alone specific sarcoma subtypes. Indeed, recent work suggests that the highest quintile of TMB within a specific cancer type is associated with improved outcomes following checkpoint inhibitor therapy and, following from that observation, that the TMB threshold for benefit is not absolute[53]. Because both MSI status (via MSIsensor) and TMB can be readily determined from targeted sequencing, correlative analysis of both MSI status and TMB in sarcoma immunotherapy trials on a subtype-specific basis is needed to inform our understanding.

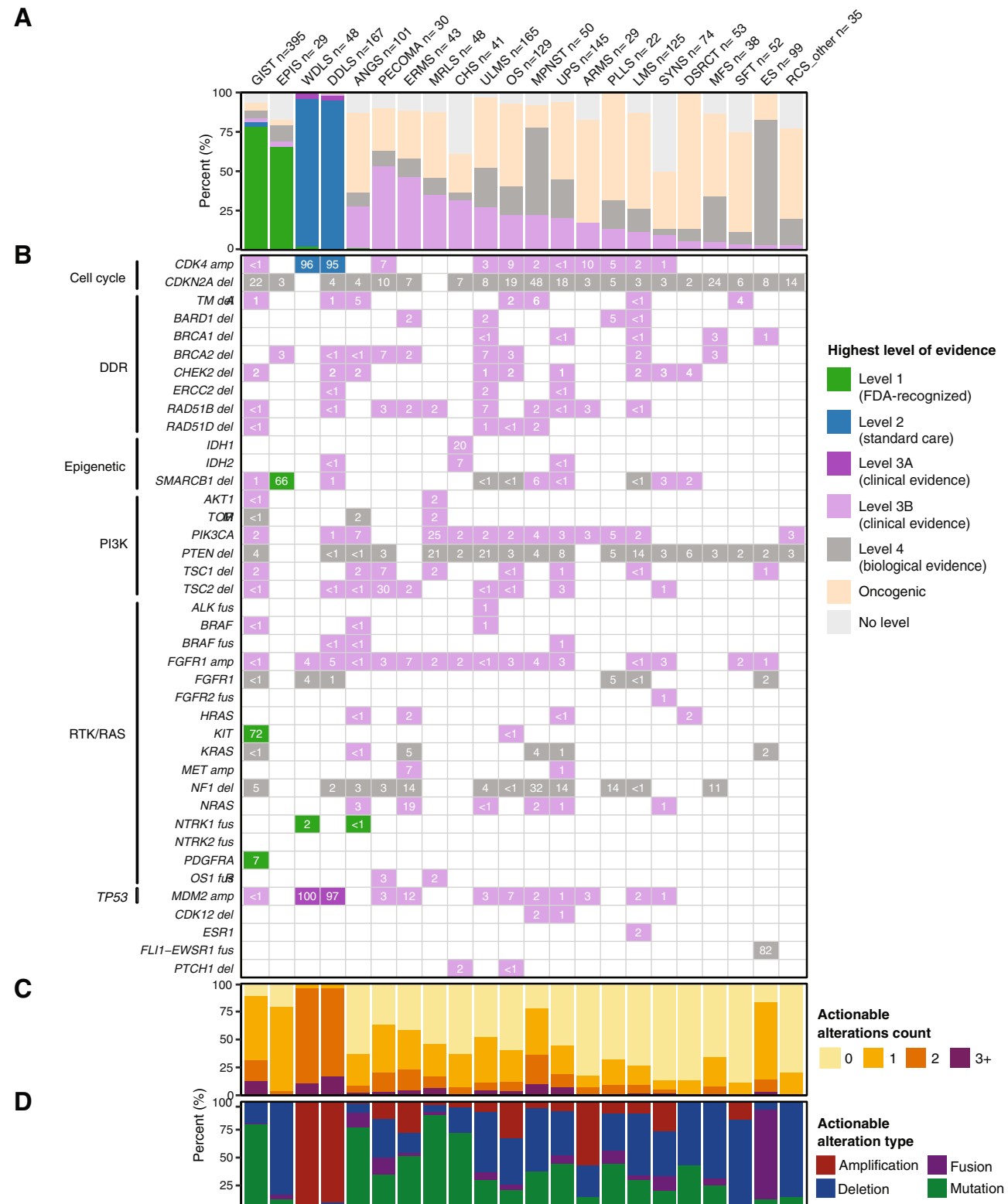

**Fig. 6 Actionability of mutations by subtype and gene.** For each of the 22 most common subtypes: **A** Frequency of actionable alterations by level of evidence. **B** Actionable alterations in individual genes, grouped by pathway. Numbers in each cell represent the percentage of samples with actionable alterations in that gene. **C** Number of actionable alterations per sample. **D** Frequency of actionable alterations classified by alteration type.

There are several important caveats and limitations inherent to the approaches we have taken in this study. The use of targeted sequencing facilitated this single-institution comparative analysis of 2138 samples and allowed for deep sequencing of genes known to be important in cancer. However, targeted sequencing is generally not well suited to identifying events such as novel genomic rearrangements, unknown drivers, or focal CNA, or specific mutational signatures such as those of impaired

homologous recombination. These features could be elucidated more effectively by whole genome sequencing (WGS), though at the cost of lower coverage. Thus, WGS will be essential to fully characterize the genetic events in this study population. In addition, while WGD estimates from MSK-IMPACT are generally concordant with those derived from whole exome sequencing[13], the WGD frequencies reported herein may be discordant from those derived by alternative analytical methods and sequencing platforms. For most of our analyses, we combined primary and metastatic samples. This choice is supported by a detailed comparison of primary vs. metastatic samples for the more common histotypes, which did not reveal any major significant differences at the genomic level, with the exception of increased TMB and FGA in metastatic vs. primary GIST samples and fewer *JUN* amplifications in metastatic vs. primary UPS samples[54]. Beyond genomic studies, there is also significant value to broader multiomic studies that combine genomic, epigenomic, and transcriptomic analysis and would extend our observations when applied to one or more specific subtypes. Other limitations stem from characteristics intrinsic to the samples, including tumor purity, which was lower in ANGS (majority of samples <50%) than other subtypes in the study, as well as limited racial diversity in the population studied.

Determining the clinical relevance of the landscape of genetic alterations in sarcomas described herein requires a further phase of investigation. Toward improved designs of clinical trials in sarcoma, which have often grouped multiple subtypes together despite significant inter- and intra-subtype genetic variability, future studies should investigate which genetic alterations result in functional effects. Indeed, the unsupervised clustering analysis we present herein (Fig. 5C, D) demonstrates that in some cases dominant genetic events (e.g. *TERT* alterations) unite distinct histologic entities. In addition, common subtypes such as UPS, LMS, ULMS, and OS can each be subclassified into multiple distinct genetic subtypes as indicated by their high entropy scores. These findings provide the rationale to incorporate tumor genomic sequencing and subsequent assignment of genotype-based groups to potentially enhance our understanding of clinical trial outcomes beyond traditional subtype-based groupings. In addition, certain cluster-defining genetic events are associated with specific vulnerabilities, which provides the rationale for considering basket trials. For instance, *ATRX* loss of function events are associated with specific clusters and may confer sensitivity to ATR inhibition[55], which is under clinical investigation as a therapeutic strategy in sarcomas (NCT03718091, NCT05071209, NCT04807816). The data we present herein and via an accompanying interactive database (https://www.cbioportal.org/study/summary?id=sarcoma_mskcc_2022) will serve as a resource for the field to explore and compare subtype-specific alterations to facilitate this transition in approach.

## Methods

**Patient cohort**. The prospective observational cohort study of tumor evaluation by MSK-IMPACT (NCT01775072) was approved by the Institutional Review Board at Memorial Sloan Kettering Cancer Center (MSK). Patients provided written informed consent to the use of their genomic data for research and were not compensated for participation. The primary outcome of this study was to determine the frequency of actionable oncogenic mutations; secondary outcomes included determining the impact of molecular profiling results on patient treatment. We identified patients with a diagnosis of soft tissue or bone sarcoma who had tumor and matched normal (usually white blood cell) tissue sequenced using the MSK-IMPACT assay through December 19, 2019[6,22]. Tumors were sequenced using one of 3 versions of MSK-IMPACT, including 341 ($n = 209$, 9.8% of samples), 410 ($n = 573$, 26.8% of samples), or 468 genes ($n = 1356$, 63.4% of samples), with results reported in the medical record. In patients with multiple samples, only one sample was included in the cohort; those collected earliest, of highest purity, and highest average coverage were selected in that order of priority. Samples sequenced earlier in the course of a patient's management were prioritized to reduce the potential influence of treatment-induced genetic changes, as these are

more likely to be collected from patients who have received fewer lines of therapy. Clinical characteristics such as patient age, sex, self-reported race, and metastatic versus primary site, were annotated per the standard MSK-IMPACT workflow[22].

**Histologic analysis**. Histologic diagnosis was annotated according to the standard MSK-IMPACT workflow. In the case of sarcomas characterized by canonical fusion events, the medical record was queried to ensure that the appropriate fusion event was detected and if not, the sample was reviewed with the assistance of an expert sarcoma pathologist considering available data from the medical record including clinical features, morphologic, and molecular analysis. Similarly, samples harboring a canonical fusion but with a discordant pathologic diagnosis were further reviewed to assign the most appropriate diagnosis. Fusions that were part of the medical record but detected by methods other than MSK-IMPACT (e.g. FISH or RT-PCR) (Supplementary Data 4) were annotated at the patient (not sample) level. In addition, samples with an ambiguous originally annotated diagnosis, including sarcoma or round cell sarcoma not otherwise specified, rhabdomyosarcoma (without further classification), spindle cell rhabdomyosarcoma, and fibrosarcoma, underwent additional medical record review and, in some cases, pathology review to render the most accurate diagnosis possible. In some additional cases with ambiguity in subtype assignment, the diagnosis was updated upon further review by an expert pathologist. We further standardized diagnoses by mapping each tumor to a unique code from the OncoTree ontology[56] except for round cell sarcoma other (RCS (other)) and extraskeletal osteosarcoma, which were categories created for this study. Samples that could not be assigned to one of the Oncotree codes ($n = 243$) were excluded from our analysis cohort. There were no samples with an assigned diagnosis of sarcoma not otherwise specified, round cell sarcoma not otherwise specified, rhabdomyosarcoma (without further classification), or fibrosarcoma included in the final cohort.

**Targeted DNA sequencing using MSK-IMPACT**. Sequencing was performed using MSK-IMPACT, a hybridization capture-based next-generation sequencing assay[6], in a Clinical Laboratory Improvement Amendments (CLIA)-certified molecular laboratory. Genomic DNA from formalin-fixed paraffin-embedded (FFPE) primary or metastatic sarcomas and patient-matched normal samples was extracted and sheared, and custom probes were synthesized for targeted sequencing of all exons and selected introns of 341, 410 or 468 genes[6,22]. Pooled libraries containing captured DNA fragments were sequenced using the Illumina HiSeq 2500 to high, uniform coverage (×>500 median coverage). All classes of genomic alterations including substitutions, indels, copy number alterations, and rearrangements were determined and called against the patient's matched normal sample. The computational pipelines for variant calling are based on standard best practices using a combination of open-source and custom-written scripts and programs[6,22].

**Computational genomic analysis**. Genomic alterations were annotated using the OncoKB precision oncology knowledge base, which identifies functionally relevant cancer variants and their potential clinical actionability[40]. Except where otherwise specified in the text, variants of unknown significance (VUS), defined as alterations not classified as *oncogenic*, *likely oncogenic*, or *predicted oncogenic* by OncoKB, were excluded from the analysis. All reported alteration frequencies were adjusted to account for the specific set of genes included in each version of the MSK-IMPACT panel by dividing the number of gene-specific alterations by the number of samples for which a given gene was sequenced. Therapeutically targetable somatic alterations were labeled using levels of clinical actionability defined in OncoKB, which range from Level 1, FDA-recognized biomarkers of response to FDA-approved drugs, to Level 4, biomarkers of hypothetical relevance based on compelling preclinical biological evidence. Analyses of alterations in oncogenic signaling pathways were performed using the set of pathway definitions previously curated by our group, which we expanded to include the DDR and epigenetic modifier pathways using additional templates curated from the literature[28,57–59].

Tumor mutation burden (TMB) was computed as the total number of nonsynonymous mutations divided by the total number of base pairs sequenced per sample. The fraction of genome altered (FGA) was defined as the fraction of genome with $\log_2$ copy number gain >0.2 or loss < −0.2 relative to the size of the genome for which copy number was profiled. We computed MSIsensor scores for all samples in the cohort using a threshold of MSIsensor score ≥10 to identify tumors with microsatellite instability (MSI-high)[60]. MSI-high was confirmed by a PCR-based assay (Idylla). MSIsensor ≥3 and <10 were labeled indeterminate and samples that did not meet quality control for assigning MSI status were labeled do not report (DNR).

Allele-specific copy number estimates at both the gene and chromosome arm levels were computed using the FACETS (Fraction and Allele-Specific Copy Number Estimates from Tumor Sequencing) algorithm, which also provided purity-corrected segmentation files and allowed identification of whole-genome duplication events[61]. FACETS output was also used to infer the cancer cell fraction associated with individual mutations for clonality analyses. Significantly recurrently mutated genes were identified using the MuSic and MutSigCV 1.4 algorithms, with a threshold q-value of 0.1[62,63].

Clustering analysis was performed as follows. All mutations, fusions, and copy number alterations were filtered for functional relevance using OncoKB. These oncogenic alterations were then aggregated into binary matrix format per gene for each patient and filtered using the 341-gene list on the IMPACT panel to generate the clustering input. Input matrix dimensionality was reduced using Uniform Manifold Approximation and Projection (UMAP) (http://arxiv.org/abs/1802.03426) via the R package umap. Clustering was performed using the Hierarchical Density-Based Spatial Clustering of Applications with Noise (HDBSCAN) method[64] via the R package dbscan[65]. All samples labelled NA (cluster 0) were unassignable to a cluster. Shannon entropy was calculated from observed cluster assignment by subtype and reported in natural units.

Mutational signatures for samples with ≥15 synonymous and nonsynonymous single nucleotide variants (SNVs) were extracted using the COSMIC v3 catalog of exome reference signatures and default parameters[66] (https://github.com/mskcc/tempoSig). This threshold was selected based on the previously reported formula[22]. For mutational signatures to be considered detectable, we required a $p$-value $< 0.05$ and a minimum of 1 observed mutation attributed to the signature, where the number of observed mutations was defined as the observed mutational signature fraction multiplied by the number of SNVs per sample.

All statistical analyses were performed using R v3.5.2 (www.R-project.org) and Bioconductor v3.4[67].

**Reporting summary**. Further information on research design is available in the Nature Research Reporting Summary linked to this article.

## Data availability

All clinical and genomic sequencing data described in this manuscript have been deposited in the cBioPortal for Cancer Genomics (PMIDs: 23550210 and 22588877) and are publicly available for online browsing and bulk download through the following link: https://www.cbioportal.org/study/summary?id=sarcoma_mskcc_2022. The raw sequencing data are protected; de-identified data are available under restricted access to protect patient privacy in accordance with federal and state law. These data can be requested for research use from the corresponding author. Data will be shared for a span of 2 years within 2 weeks of execution of a data transfer agreement with MSK, which will retain all title and rights to the data and results from their use. Data on individual patients and gene fusions are also provided in Supplementary Data and Supplementary Tables. TCGA data used for comparison is available via the Genomic Data Commons Portal: https://portal.gdc.cancer.gov/.

## Code availability

Custom code used for analyses is publicly accessible (https://github.com/mskcc).

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

## Acknowledgements

This study was supported by the NCI MSK SPORE in Soft Tissue Sarcoma P50 CA217694 (SS), the Marie-Josée and Henry R. Kravis Center for Molecular Oncology (DBS), Cycle for Survival (PC, WDT), NCI K08 CA245212 (BAN), NCI R01 CA228216 (PC), Orphan Products Grants Program/U.S. Food and Drug Administration R01 FD005731 (PC), and the Geoffrey Beene Cancer Research Fund (PC). The Memorial Sloan Kettering Cancer Center Support Grant (P30 CA008748) supported core facility resources used in this research. The manuscript was edited by Jessica Moore (MSK Editorial and Grant Services).

## Author contributions

Conception and design: B.A.N., F.S-V., M.M.G., T.G.B., S.S., and W.D.T. Development of methodology: B.A.N., F.S-V., S.A.S., H.S., C.T., and N.D.S. Data collection: B.A.N., C.R.A., A.Z., M.M.G., P.C., S.P.D'A, M.A.D., G.M.K., M.L.K., S.M., K.T., P.A.M., L.H.W., E.K.S., J.L.G.B., N.N.S., M.L.H., J.H.H., L. Q., A.M.C., S.S.Y., B.R.U., S.C., N.P.A., M.R.H., M.F.B., D.B.S., M.L., S.S., and W.D.T. Data curation: B.A.N., F.S-V., S.A.S., C.R.A., and T.G.B. Resources: F.S-V., S.A.S., H.S., C.T., A.Z., M.F.B., and N.S. Analysis and interpretation of data: B.A.N., F.S-V., S.A.S., C.R.A., E.R., H.S., C.T., N.D.S., S.R., R.G-M., A.Z., A.L., B.J., A.O. J.A.S., J.S., M.F.B., D.B.S., N.S., S.S., and W.D.T. Drafting the manuscript: B.A.N., F.S-V., E.R., S.S., and W.D.T. Review and revision of the manuscript: All authors, including Chan and Alektiar. Visualization of data: F.S-V., S.A.S., H.S., and C.T. Study supervision and funding acquisition: S.S. and W.D.T.

## Competing interests

M.M.G. has served on advisory boards for Athenex, Ayala, Bayer, Boehringer Ingelheim, Daiichi Sankyo, Epizyme, Karyopharm, Rain, SpringWorks Therapeutics, Tracon, and TYME Technologies; provides consulting services through Guidepoint, GLG Pharma, Third Bridge, and Flatiron Health; has received speaking honoraria from Medscape, More Health, Physicians Education Resource and touchIME; receives publishing royalties from Wolters Kluwer; holds a patent for a patient-reported outcome tool licensed through the institution; and has performed research without compensation in collaboration with Foundation Medicine. T.G.B. is currently employed by Pfizer. P.C. has served on advisory boards or consulted for Deciphera, Exelixis, NingboNewBay Medical Technology, Novartis, and Zai Lab, and has received institutional research funding from Deciphera, Ningbo NewBay Medical Technology, Novartis, and Pfizer/Array. S.P.D. has received institutional research funding from Amgen, Bristol Meyers Squibb, Deciphera, EMD Serono, Incyte, Merck, and Nektar Therapeutics, has served as a consultant or on advisory boards for Adaptimmune, Amgen, EMD Serono, GlaxoSmithKline, Immune Design, Immunocore, Incyte, Merck, and Nektar Therapeutics, Pfizer, Servier, and Rain Therapeutics, and has served on data safety monitoring boards for Adaptimmune, GlaxoSmithKline, Merck, and Nektar Therapeutics. M.A.D. has received institutional research funding from Aadi Bioscience and Eli Lilly. C.M.K. has received research funding from Amgen, Exicure, Incyte, Kartos, Merck, Servier, and Xencor; consulted for Chemocentryx and Kartos; served on a data safety review board for Kartos; and served on advisory boards for Immunicum. S.M. has received research funding from Ascentage Pharma and Hutchison Medi Pharma. K.T. has served as a consultant for Epizyme and GlaxoSmithKline. P.A.M. has served on advisory boards or consulted for Margaux Miracle Foundation, Salarius Pharmaceuticals, and Takeda, and has an immediate family member who has served on advisory boards or consulted for Boehringer Ingelheim and Genentech and received honoraria from Eastern Pulmonary Conference. E.K.S. has consulted for Epizyme. J.L.G.B. has received institutional research support from Amgen, Bayer, Bristol Myers Squibb, Celgene, Cellectar Biosciences, Eisai, Ignyta, Lilly, Loxo Oncology, Merck, Novartis, and Roche; and served on data safety monitoring boards for Abbvie, Merck, and SpringWorks and on an advisory board for Bristol Myers Squibb. M.L.H. has served on advisory boards and consulted for Eli Lilly, GlaxoSmithKline, and Thrive Bioscience, received author royalties from UpToDate, and received speaker honoraria from Research to Practice; her spouse is employed by Sanofi. J.H.H. has consulted for Daiichi Sankyo and Stryker and is a trustee of the Musculoskeletal Transplant Foundation and board member for the Make It Better Foundation to Cure Childhood Osteosarcoma. A.M.C. has served on an advisory board for SpringWorks. B.R.U. is co-inventor of intellectual property (H.R.A.S. as a biomarker of tipifarnib efficacy) that has been licensed by MSK to Kura Oncology. S.C. has consulted for AstraZeneca. M.F.B. has served as a consultant for Eli Lilly and PetDx. W.D.T. has consulted for Adcendo, Amgen, AmMax Bio, Ayala Pharmaceuticals, Bayer, Blueprint, C4 Therapeutics, Cogent, Daiichi Sankyo, Deciphera, Eli Lilly, EMD Serono, Epizyme, Foghorn Therapeutics, Kowa, Medpacto, Mundipharma, and Servier; served on advisory boards for Certis Oncology and Innova Therapeutics; holds two patents for biomarkers of CDK4 inhibitor efficacy in cancer, and is a co-founder of and owns stock in Atropos Therapeutics. D.B.S. has consulted for BridgeBio, FORE Therapeutics, Loxo/Lilly Oncology, Pfizer, Scorpion Therapeutics, and Vividion Therapeutics. All other authors have no competing relationships to disclose.

## Additional information

[1]Department of Medicine, Memorial Sloan Kettering Cancer Center, New York 10065 NY, USA. [2]Department of Medicine, Weill Cornell Medical College, New York 10065 NY, USA. [3]The Laboratory of Chromatin Biology and Epigenetics, The Rockefeller University, New York 10065 NY, USA . [4]Department of Surgery, Memorial Sloan Kettering Cancer Center, New York 10065 NY, USA. [5]Marie-Josée and Henry R. Kravis Center for Molecular Oncology, Memorial Sloan Kettering Cancer Center, New York 10065 NY, USA. [6]Department of Pathology, Memorial Sloan Kettering Cancer Center, New York 10065 NY, USA. [7]Department of Epidemiology and Biostatistics, Memorial Sloan Kettering Cancer Center, New York 10065 NY, USA. [8]Physiology, Biophysics and Systems Biology Graduate Program, Weill Cornell Medical College, New York 10065 NY, USA. [9]Bioinformatics Core, Memorial Sloan Kettering Cancer Center, New York 10065 NY, USA. [10]Human Oncology and Pathogenesis Program, Memorial Sloan Kettering Cancer Center, New York 10065 NY, USA. [11]Department of Pediatrics, Memorial Sloan Kettering Cancer Center, New York 10065 NY, USA. [12]Department of Surgery, Weill Cornell Medical College, New York 10065 NY, USA. [13]Department of Radiation Oncology, Memorial Sloan Kettering Cancer Center, New York 10065 NY, USA. [14]Present address: Department of Epidemiology and Biostatistics, Memorial Sloan Kettering Cancer Center, New York 10065 NY, USA. [15]These authors contributed equally: Benjamin A. Nacev, Francisco Sanchez-Vega. ✉email: singers@mskcc.org; tapw@mskcc.org

