## [Peer Review File · Nature Communications]

Clinical sequencing of soft tissue and bone sarcomas delineates diverse genomic landscapes and potential therapeutic targetsEditorial Note: Parts of this Peer Review File have been redacted as indicated to maintain the confidentiality of unpublished data.

REVIEWER COMMENTS

Reviewer #1 (Remarks to the Author): Expert in sarcoma clinical research and genomics

In the manuscript entitled “Clinical sequencing of soft tissue and bone sarcomas delineates diverse genomic landscapes and potential therapeutic targets”, Nacev and colleagues provide data on panel sequencing of 2,138 soft tissue sarcomas, representing 45 pathological entities seen in an expert center. The data generated is based on the MSKCC impact panel which profiles 341, 410 or 468 sarcoma specific genes and notably is FDA approved.

Certainly, the authors are to be commended for the sheer output of this study. The detailed clinical outcomes coupled with expert pathology review are welcome real time data for clinical experts in the fields. The manuscript in its current format, however, is limited in its scope to truly understand potential new targets and does not direct the reader in how the data is informative to define targeted therapies in the era of precision medicine.

The following major comments warrant further clarification:

1. Overall, the technology used in this manuscript does not refine the state-of-the-art molecular analysis of sarcomas that have been recently published using WGS, methylation assays and/or comprehensive transcriptomes. The authors should tempter the translation of this manuscript in the introduction (lines 31-35), where more data is assumed to be able to “facilitate diagnostic precision, identify biomarkers etc....”. Panel based sequencing lacks the refinement to more accurately understand the molecular mechanisms that drive genomic instability, fusion formation, and the ability to accurately define mutational signatures. It is unclear how many patients were in each of the respective version of the panel as the number of genes being examined expanded over the course of the study (341, 410 or 468 genes of MSK-IMPACT).
2. The mutational frequencies of TP53 and RB1 have been variability reported in the sarcoma literature to date (PMID: 30889380 and 29100075). With this manuscript the frequencies are seen here in key genes known in sarcoma seen in Figure 2B, how does this compare to WGS recently published? Why there are discrepancy in rates/detection arises should be discussed (tumor purity, technology etc). This does not negate the use of panel sequencing, yet merely provides the context in which this technology is operative, i.e. RB1 is merely 19% in LMS yet with WGS has been suggested to be almost universal.
3. The results section lacks a higher order interpretation of the data the subheading “Sample and patient characteristics” is generic and does not convey what is critical about the dataset that has been amassed in this manuscript (also Figure 1 label). Another example is “Copy number alterations by subtype.” Also, commonly disrupted pathways in Sarcoma” and subheading, “Mutual exclusivity and co-occurrence” of ? These generic headings convey a lack of novelty.
4. How were VUS interpreted? (lines 90-94). Were genetic syndromes – LFS, NF1 accounted for in the cohort? Please provide this data if available.
5. What % of ANGS were radiation associated vs. sporadic? (Lines 183-4)” Given the association of ANGS with prior ionizing radiation, ATM mutations may represent a convergent pathogenic mechanism for accumulation of DNA damage.” What is meant by the previous phrase? It is vague. Also, what was the PI3K mutational spectrum in ANGS?
6. Lines 112-120 are not particularly novel data with respect to LPS biology. (PMID: 29100075, 31831742 and 25517748). Certainly, cell cycle dysregulation should be found given that it is

diagnostic for WD/DDLPS and is the molecular rationale for the use of CDK4i. What are novel findings about LPS mutations found in this particular study?

7. How does understanding frequency and/or timing of WGD inform patient care? Appreciate Figure 3C and 3D, however data as presented is an association. The rationale of why WGD may impact outcome isn't articulated in the current manuscript which is a missed opportunity.

8. Figure 5 is an exciting data set which might inform the rationale for basket trials. How is Figure 5C be linked to current – upcoming clinical trials? This section warrants expansion and would add novelty to the manuscript. For example, emerging compounds to target ATRX.

9. Are there any biomarkers to predict why there are higher TMB seen in subtypes that have shown promise in previous I/O trials?

10. Please resolve how mutational signatures such as HRD in LMS are not found in this study (PMID: 29321523 and 343301934)? Why was a threshold of ≥ 15 SNV selected? Did the clock-like signature internally validate data set? This section is not particularly novel as it validates previously published datasets (PMID: 32042194 and 33016928). Concern that the methods used do not provide adequate resolution to accurately generate mutational signatures.

11. Although the panel can provide data on MSI status, would be appropriate to justify why this would be informative in sarcoma as in this dataset and others, MSI is not considered a major DNA damage pathway in this disease. Was there any data to suggest NHEJ dysfunction or HRD? Why not use HRD detect (PMID: 33283135)? It is critical to develop classifiers and tools that can be used for discovery and/or inform mesenchymal biology. Suggest Figure 6 should be supplemental.

Additionally, the following minor comments should be addressed:

1. Line 16 states sarcomas arise in “organ sites”. Most sarcomas do not arise in organs per se, please clarify and suggest coupling with phrase “connective tissue”. Also, in Figure 1A please define the nomenclature of retroperitoneal vs. retro/IA? How does this distinction help in prognostication?

2. In Line 18 please provide context for studying pediatric sarcomas, as they are in fact a common solid tumor in this age group. Does this manuscript seek to address molecular signatures across age span? Also, TCGA, which is foundational for the panel design didn't evaluate pediatric sarcoma subtypes. Does this impact the utility of sequencing of pediatric subtypes that MSK-Impact is now being applied to?

3. The authors are to be commended for the detailed clinicopathologic data, especially survival outcomes provided in Figure 1. However, in some subtypes (ANGS) tumor purity is $<50\%$ in the majority of samples. Also, with respect to EDI, race does not appear to be a representative demographic of the US population, does this limit the interpretation and translatability?

4. In Figure 2, it is of concern that the pathognomonic fusion for ARMS isn't found in the majority of specimens (? 30-40%), please clarify this why this wasn't the case. In the methods section it is commented that if there was a discrepancy in diagnosis this was reviewed by a sarcoma expert and reassigned. Please provide this data as would be informative in how this workflow occurs in a precision medicine program.

5. Please clarify nomenclature of ULMS vs LMS this is confusing at times. Please provide data in figure 5A for LMS as only ULMS is represented.

6. Supplemental Figure 1B label needs to be corrected.

Reviewer #2 (Remarks to the Author): Expert in bioinformatics, sarcoma genomics and subtypes

Singer et al. propose a unique pan-sarcoma mutational landscape of 22 sarcoma types based on 1918 samples profiled on the MSK-IMPACT targeted gene panels.

They first describe the cohort and cohort-level clinical features, including purity, sequencing depth, age and survival; then, the number and type of driver alterations detected, pointing to clustered FLT4 VUS missense mutations in angiosarcomas; then copy-number aberrations (CNA), the prevalence of CNA events and whole genome doubling (WGD), relating WGD status to survival in metastasis of UPS; gene-level and pathway-level mutation frequencies across sarcoma types, with interesting high-prevalence PI3K, TERT and epigenetic-related alterations highlighted; they look at gene-level and pathway-level co-occurrence and mutual exclusivity of alterations, with a detailed look at the common ATRX mutations; and propose a summary view across types by clustering samples by their alteration profiles; they go on to characterise MSI and TMB, two clinically relevant features, with mutational signature analysis showing UV signature in high TMB samples; and finally look at the clinical actionability of the mutations.

Their data release accompanying the manuscript would also include data for an additional 220 samples for other underrepresented sarcoma types, not analysed here.

This is an important resource for the field, first of this kind in sarcoma genomics. The manuscript is rich in details and the data is well-presented and well-described. It offers interesting parallels and distinctions across these 22 different sarcoma types and a "global view" based on their genomic alterations as well as pointers for clinical management.

We have a few comments that we hope could improve this manuscript.

Comments:

* It is important that this dataset is released with the manuscript. The data has not been made available to reviewers but there is a promise of "bulk download" of genomic and clinical data through the cBioPortal website. What data and in which format it will be available could be made clearer.

* Metastases and primary samples are used as a combined group. But are there differences between metastases and primary samples?

* As the authors are using data from different versions of their gene panel (with increasing number of genes), it is not clear how frequencies and counts are derived? I think absolute counts should be shown, while frequencies should be based on the number of samples for which the gene is included in the panel (not total amount of samples). Is this the case? If so, a brief description of how this is taken into account in the methods is missing.

* The variant calling strategies are not described in the methods, how is variant calling performed? Please provide at least a link to a previous paper where this is described to make the analyses interpretable/reproducible.

* In the methods, please describe exactly what data goes in the UMAP+HDBSCAN. This is one of the main pan-sarcoma result, but it is not clear how it was derived.

* In silico ploidy inference (and thus WGD status) is underdetermined and ambiguous (discussed here: <https://doi.org/10.1038/s41592-020-01013-2>), and in the presented study, WGD prevalence seems underestimated at least for UPS. Indeed, 65% TCGA samples had undergone WGD (<https://doi.org/10.1016/j.cell.2017.10.014>); in Steele et al., 90% undifferentiated sarcomas showed WGD with experimental validation of ploidies (<https://doi.org/10.1016/j.ccell.2019.02.002>). We would

expect even higher rates in the dataset of Singer et al., as >30% cases are metastatic samples, expected to present with even higher prevalence of WGD. Because of the limited genome resolution of these targeted panels, perhaps FACETS would favour low-ploidy solutions; in any case, the ploidy and WGD status should be interpreted with a grain of salt, i.e. an appropriate discussion would be useful.

* In general, the pros and cons of the targeted-panel approach could be discussed better in the context of this pan-sarcoma genomics study. The identification of important genetic changes, i.e. focal CNA, fusions, ploidy changes/WGD status, unknown driver genes is naturally limited. Pan-sarcoma whole-genome sequencing studies are needed to identify more events that are not captured here. Though maybe trivial, this should be mentioned and discussed by the authors.

* The rationale behind the selection of a single representative sample in the multi-sample cases (= early date, purity, coverage) is not really explained. Why does it make sense in that order? Instead why not e.g. pick primary site over metastasis, then the highest number of reads per tumor chromosomal copies (compound power metric based on copy-number, read depth and purity, please see: <https://doi.org/10.1016/j.cell.2021.03.009>, <https://doi.org/10.1038/s41592-020-01013-2>), then select the largest gene panel (=widest genomic coverage)?

* This paper is a unique pan-sarcoma genomics paper, as it covers so many sarcoma types. But therefore, the underrepresented sarcoma types (220 samples, an extra 11%) should be included in the analyses. It is a big loss not to have them shown here. Especially, in the summary UMAP results, it might show where these other sarcoma types sit relative to the 22 analysed. If no strong signal comes out, they could be pooled in a category "Others" for most of the other figures.

* Figure 1: avoid loaded term of "Race"? Perhaps replace by "Population"?

* Supp Fig 4B: x-axis labels read "Copy number slteration"

REVIEWER COMMENTS

Reviewer #1 (Remarks to the Author): Expert in sarcoma clinical research and genomics

In the manuscript entitled “Clinical sequencing of soft tissue and bone sarcomas delineates diverse genomic landscapes and potential therapeutic targets”, Nacev and colleagues provide data on panel sequencing of 2,138 soft tissue sarcomas, representing 45 pathological entities seen in an expert center. The data generated is based on the MSKCC IMPACT panel, which profiles 341, 410, or 468 sarcoma-specific genes and notably is FDA-approved.

Certainly, the authors are to be commended for the sheer output of this study. The detailed clinical outcomes coupled with expert pathology review are welcome real-time data for clinical experts in the fields. The manuscript in its current format, however, is limited in its scope to truly understand potential new targets and does not direct the reader on how the data is informative to define targeted therapies in the era of precision medicine.

We thank the reviewer for this summary and are pleased that the data are considered of potential benefit to the field. We agree with the comment that the lack of novel target definition in this study limits its scope, but such an outcome would require experimental models and extensive preclinical evaluation and is beyond the scope of our project. Rather, our aim was to provide a genomic analysis of a large and unique breadth of sarcoma subtypes to compare and identify genetic events that will be of interest and use to a broad sarcoma community. This community includes clinical experts, as highlighted by the reviewer, as well as basic scientists who have an interest in studying the fundamental genetic events in multiple rare sarcoma subtypes.

The following major comments warrant further clarification:

1. Overall, the technology used in this manuscript does not refine the state-of-the-art molecular analysis of sarcomas that have been recently published using WGS, methylation assays, and/or comprehensive transcriptomes. The authors should temper the translation of this manuscript in the introduction (lines 31-35), where more data is assumed to be able to “facilitate diagnostic precision, identify biomarkers, etc....”.

We agree with the reviewer that these data and the analysis and conclusions of this manuscript should be considered part of a larger multidisciplinary and comprehensive effort needed to define and understand the genetic, epigenomic, and gene expression states that define myriad sarcomas. Applications of WGS, DNA methylation, and transcriptomic studies as referenced by the reviewer will of course also be essential aspects of this process. We have therefore updated the text in the area highlighted by the reviewer to reflect this (lines 33–38):

“Analysis of a larger cohort could define the frequency of potentially actionable alterations in rare sarcoma subtypes and broadly compare the frequency of genetic alterations across subtypes. These data, when integrated with other ‘multiomic’ sarcoma studies, will facilitate better diagnostic precision, identify prognostic biomarkers, improve laboratory-based modeling of sarcomas, and generate novel hypotheses on underlying disease mechanisms.”

We have also added a mention of the value of ‘multiomic’ studies and the need to integrate our findings with others’ to the new limitations paragraph of the Discussion (lines 447–450).

We respectfully disagree that “...the technology used in this manuscript does not refine the state-of-the-art molecular analysis of sarcomas...”. As discussed in the manuscript, an important value of this work is that it provides an analysis of 45 different subtypes on a uniform platform with germline controls within a single resource, including very rare entities. This allows comparisons between histologic subtypes of sarcomas and facilitates downstream analysis such as the unsupervised clustering of sarcoma subtypes based on mutational profiles (Fig. 5C,D) and analysis of actionable mutations (original Fig. 7 [now Fig. 6]).

Panel-based sequencing lacks the refinement to more accurately understand the molecular mechanisms that drive genomic instability and fusion formation, and to accurately define mutational signatures.

We agree that the use of panel-based sequencing is a limitation of our study, particularly in terms of identifying novel gene fusion and copy number events, and we have now acknowledged this in the Discussion (lines 432–441). While whole-exome sequencing (WES) and whole-genome sequencing (WGS) are not yet widely used in routine clinical practice, we share the hope that they will be useful in future clinical sequencing studies and will enhance our understanding of the molecular mechanisms driving sarcoma tumorigenesis. We additionally note:

1. TMB estimates derived from MSK-IMPACT correlate highly with estimates obtained through WES recapture of the same samples (Zehir et al. Nat Med 2017; Rizvi et al. J Clin Oncol 2018), and with WGD calls (Bielski et al. Nat Genet 2018). [REDACTED]

[REDACTED]

2. The MSK-IMPACT assay has been successfully used to analyze mutational signatures in the past (both generally across cancer types (Zehir et al. Nat Med 2017), and within very specific molecular subtypes (Yang et al. Mod Pathol 2020; Caso et al. J Thorac Oncol 2020). It is true that panel-based methods work better for detection of signatures with highly distinct profiles, such as UV light or smoking signatures, while other signatures such as those associated with homologous recombination deficiency may be better studied using alternative sequencing platforms with higher breadth of coverage (Jonsson et al., Nature 2019). Nonetheless, our mutational signature analysis is hypothesis-generating and we believe that it will be informative and useful for interested readers.
3. Using alternative approaches such as WES or WGS instead of MSK-IMPACT would require lower sequencing depths, which in turn might result in some relevant mutations being missed in important cancer genes. When MSK-IMPACT data was downsampled from its original ~800x mean target depth to a lower mean target depth of 150x, which is typical of WES studies, ≥ 9% of all mutations would have been missed, including therapeutically targetable alterations in *BRAF*, *EGFR*, and *MET* (Zehir et al., Nat Med 2017). Thus, using those alternative methods would have both advantages and disadvantages.

It is unclear how many patients were in each of the respective versions of the panel as the number of genes being examined expanded over the course of the study (341, 410 or 468 genes of MSK-IMPACT).

We thank the reviewer for raising this important point. The exact version of the panel used to sequence each individual sample is provided in Supplementary Table 1. In our revised manuscript, we now also provide overall counts for each panel in the following sentence in the Patient Cohort subsection of the Methods (lines 477–479):

“Tumors were sequenced using one of 3 versions of MSK-IMPACT, including 341 (n=209, 9.8% of samples), 410 (n=573, 26.8% of samples), or 468 genes (n=1356, 63.4% of samples)”.

2. The mutational frequencies of *TP53* and *RB1* have been variably reported in the sarcoma literature to date (PMID: 30889380 and 29100075). With this manuscript the frequencies are seen here in key genes known in sarcoma seen in Figure 2B. How does this compare to WGS recently published? Why discrepancies in rates/detection arise should be discussed in terms of tumor purity, technology, etc. This does not negate the use of panel sequencing, yet merely provides the context in which this technology is operative, i.e. the rate of *RB1* alterations is merely 19% in LMS in this study, yet with WGS has been suggested to be almost universal.

We thank the reviewer for raising this point. As the reviewer mentions, a number of factors could explain the discrepancies in the reported frequencies of gene alterations. For example, the WGS approach used by in the Steele et al. reference uses a depth of sequencing of 70x, while MSK-IMPACT uses ~800x on average, so certain mutations (e.g. those with lower allele frequency) could be detected by MSK-IMPACT but not by WGS. On the other hand, the WGS platform is better designed to identify structural rearrangements, such as novel translocations, thanks to its higher breadth of coverage, so certain alterations could be detected by WGS but not MSK-IMPACT (in fact, Steele et al. mention that “that up to 50% of driver events in *TP53*, *RB1*, and *ATRX* would have been missed if only exome data were available for the cohort”). There may also be differences due to the different annotation pipelines used to distinguish drivers from variants of unknown significance, etc.

For *TP53*, the rate of alterations in LMS reported by the TCGA study is almost identical to the rate observed in our study (62% vs. 63%, Figure R2). For *RB1*, the rate of reliably called alterations in the LMS set reported by TCGA is actually only 30%, including 11% altered by deep deletions and 19% affected by mutations (Figure R2); the remaining alterations were shallow deletions, affecting 78% of the TCGA samples. Shallow deletions are more susceptible to noise and less likely to correlate with actual levels of gene expression in our analyses, and were thus excluded. The overall rate of *RB1* alterations in LMS, including deep deletions and mutations, in our study is actually 41% (Figure 4A), 10% higher than reported by TCGA. Of note, the 19% rate of *RB1* mutations in the TCGA study is exactly the same as the rate of mutations we report (Figure 2B); we find a greater rate of deep deletions, which may be due to the use of different technologies and computational pipelines (SNP6 vs. targeted sequencing) for calling these gene-level events.

Figure R2. Comparison of *TP53* and *RB1* alterations in LMS between the TCGA study and our manuscript (MSK-IMPACT).

3. The results section lacks a higher order interpretation of the data. The subheading “Sample and patient characteristics” is generic and does not convey what is critical about the dataset that has been amassed in this manuscript (also Figure 1 label). Another example is “Copy number alterations by subtype.” Also, commonly disrupted pathways in Sarcoma” and subheading, “Mutual exclusivity and co-occurrence” of ? These generic headings convey a lack of novelty.

As suggested, we have revised many of the results section headings and the Figure 1 label to better convey the major findings of the section. Please see highlighted changes in the revised manuscript.

4. How were VUS interpreted? (lines 90-94).

VUS were annotated using OncoKB, which classifies each somatic alteration according to its known or suspected biological relevance (oncogenic, likely oncogenic, predicted oncogenic, likely neutral, or unknown oncogenicity) and effect (gain-of-function, loss-of-function, switch-of-function or neomorphic, neutral or unknown) based on the available literature. Alterations categorized as likely neutral or with unknown oncogenicity were treated as VUS, while the rest (oncogenic, likely oncogenic or predicted oncogenic) were treated as driver alterations. A detailed description of the OncoKB annotation and interpretation pipeline has been published and is referenced in the original version of our manuscript in the “Computational Genomic Analysis” subsection of the Methods (Chakravarty et al. JCO Precis Oncol 2017). Notably, OncoKB's processes, validation, data integrity and security, and transparency have received official FDA recognition, which speaks to the reliability of this classifier.

To make this clearer to readers, we updated the “Computational Genomic Analysis” subsection of the Methods (lines 525–527):

“Except where otherwise specified in the text, variants of unknown significance (VUS), defined as alterations not classified by OncoKB as oncogenic, likely oncogenic, or predicted oncogenic in OncoKB, were excluded from the analysis.”

Were genetic syndromes such as LFS and NF1 accounted for in the cohort? Please provide this data if available.

We did not include germline variants in our analysis because our study is focused on somatic events. In addition, only a small fraction of patients gave consent for germline sequencing, which would significantly limit our sample size were we to pursue such an analysis.

5. What % of ANGS were radiation-associated vs. sporadic? (Lines 183-4) “Given the association of ANGS with prior ionizing radiation, *ATM* mutations may represent a convergent pathogenic mechanism for accumulation of DNA damage.” What is meant by the previous phrase? It is vague.

Radiation-associated versus sporadic status is not annotated in the IMPACT dataset. Respectfully, we suggest that characterizing genetic differences between these two groups of ANGS would be better suited to an independent manuscript designed to specifically study that question. Regarding the highlighted statement, we have updated the text to better express our hypothesis that accumulation of DNA damage, from ionizing radiation (a known association) and/or impaired DNA damage repair pathways (a possibility raised by our data), may be part of the common mechanism for sarcomagenesis in angiosarcoma (lines 199–203). If these edits have not clarified our intention sufficiently, we will remove this sentence from the manuscript, as it is not critical to the overall findings.

Also, what was the PI3K mutational spectrum in ANGS?

As shown in Figure 4A, the overall rate of oncogenic events in the PI3K pathway in ANGS is 16%. We have also updated Figure 4B, which displays rates of alterations of specific PI3K pathway genes, to include ANGS. The revised manuscript briefly discusses this analysis (lines 172–175).

6. Lines 112-120 are not particularly novel data with respect to LPS biology. (PMID: 29100075, 31831742 and 25517748). Certainly, cell cycle dysregulation should be found given that it is diagnostic for WD/DDLPS and is the molecular rationale for the use of CDK4i. What are novel findings about LPS mutations found in this particular study?

We completely agree with the reviewer that cell cycle events, including amplification of *CDK4* and *MDM2*, are a hallmark of WD/DDLS. To make this clear to the reader, we have revised the text to state that *CDK4* and *MDM2* amplification events are expected and have also added the 3 references suggested by the reviewer (lines 153–154).

Because, as the reviewer points out, genetic alterations in cell cycle regulators are common in WDLS and DDLS, we focused our discussion in the sections on chromosomal gains and losses (lines 121–130) and alterations in epigenetic regulators (lines 236–241) on the *differences* between WDLS and DDLS given their divergent clinical behavior in order to provide added value to field. These differences between WDLS and DDLS might generate hypotheses of interest for investigators interested in factors that mediate the increased aggressiveness, including metastatic potential and impaired differentiation, in DDLS compared to WDLS.

7. How does understanding frequency and/or timing of WGD inform patient care? Appreciate Figure 3C and 3D, however data as presented is an association. The rationale of why WGD may impact outcome isn't articulated in the current manuscript which is a missed opportunity.

We thank the reviewer for highlighting the opportunity to better convey the relevance of WGD to the reader. To address this, we have added the following to the manuscript (lines 131–136):

“We also analyzed whole genome doubling (WGD) events across subtypes and compared them to data from a pan-cancer analysis where WGD was associated with decreased overall survival (Bielski et al., Nat Genet 2018). In that study, WGD was further associated with defects in cell cycle regulation and increased proliferative rates, which could explain differences in patient outcomes. Therefore, WGD warrants further study as a potential prognostic biomarker in sarcomas on a subtype-specific basis.”

8. Figure 5 is an exciting dataset which might inform the rationale for basket trials. How is Figure 5C to be linked to current and upcoming clinical trials? This section warrants expansion and would add novelty to the manuscript. For example, emerging compounds to target ATRX.

We agree with the reviewer that incorporating genomic characterization in addition to histologic characterization as part of clinical trial design is critically important to advancing our field. Comparing genetic events between and within a wide range of sarcoma subtypes is a major goal of our study for exactly the reasons the reviewer has highlighted. To convey this point more robustly to the reader, we have significantly improved the Discussion (lines 457–468) in reference to the data presented in Figure 5:

“Indeed, the unsupervised clustering analysis we present herein (Figure 5C-D) demonstrates that in some cases dominant genetic events (e.g. TERT alterations) unite distinct histologic entities. In addition, common subtypes such as UPS, LMS, ULMS, and OS can each be subclassified into multiple distinct genetic subtypes as indicated by their high entropy scores. These findings provide the rationale to incorporate tumor genomic sequencing and subsequent assignment of genotype-based groups to potentially enhance our understanding of clinical trial outcomes beyond traditional subtype-based groupings. In addition, certain cluster-defining genetic events are associated with specific vulnerabilities, which provides the rationale for exploring basket trials. For instance, ATRX loss-of-function events are associated with specific clusters and may confer sensitivity to ATR inhibition (Flynn et al. Science 2015), which is under clinical investigation as a therapeutic strategy in sarcomas (NCT03718091, NCT05071209, NCT04807816).”

9. Are there any biomarkers to predict why higher TMB is seen in subtypes that have shown promise in previous I/O trials?

Based on a large pan-cancer cohort of 100,000 cases, TMB has been shown to correlate with alterations in approximately 250 genes (Chalmers et al., Genome Med 2017), including genes involved in the mismatch repair (MMR) pathway, among others. MMR deficiency is often associated with microsatellite instability, which we showed in our analysis was overall very low for sarcomas as expected based on previous studies. In Figure 6B, we directly compared TMB and MSI status, and found that “...while microsatellite instability corresponded with high TMB, the inverse was not true.” This result implies that there may be correlations between TMB status and one or more of the non-MMR pathway genes in the set of ~250 genes identified to correlate with TMB in the above-referenced publication. Based on our understanding of the reviewer’s

very good question, the best next step to identify predictive biomarkers for elevated TMB in one or more immunotherapy-sensitive sarcoma subtypes would be to test for association between alterations in one or more of those genes and high TMB in specific subtypes that are more likely to respond to immunotherapy (e.g. UPS, cutaneous and head and neck ANGS, ASPS). We respectfully suggest that while this is an important question, such a detailed and subtype-focused analysis is best suited for dedicated subtype-specific follow-up studies that could also leverage datasets that include response to immunotherapy.

10. Please resolve how mutational signatures such as HRD in LMS are not found in this study (PMID: 29321523 and 343301934)?

As discussed in response to a preceding comment, the mutational profile of certain signatures such as those related to HRD makes it preferable to use sequencing platforms with higher breadth of coverage, such as WES or WGS, for their analysis. This limitation is discussed in the new paragraph in the Discussion on the caveats and limitations of our study (lines 434–441). Some of the members of our team investigated this question in detail in a separate study that included 62 LMS cases and combined MSK-IMPACT sequencing with WES for a small subset of them (Jonsson et al., Nature 2019). Because we are aware of this limitation of the MSK-IMPACT platform, and because we do not have WES or WGS for the cases analyzed in our current study, we decided not to discuss HRD signatures in the present manuscript.

Why was a threshold of ≥ 15 SNV selected?

We chose a threshold of ≥ 15 SNVs for mutational signature analyses based on the analysis of the first 10,000 cases analyzed with MSK-IMPACT (Zehir et al. Nat Med 2017). In that manuscript, the authors analyzed the overall TMB distribution over the entire cohort of 10,000 samples and concluded that an adequate number of mutations for analysis of mutational signatures was given by the median TMB + $2 \times \text{IQR}(\text{TMB})$, where IQR is the interquartile range. In that case, this formula gave an exact value of 13.8 mutations/Mb. This number corresponds to a minimum of 14–15 mutations, depending on the version of the MSK-IMPACT assay used for each sample (different versions of the panel have different coverage). Because of this, to be conservative with our conclusions, we decided to use a threshold of 15 SNVs for the present study.

Did the clock-like signature internally validate the data set?

Validating the clock-like signature is not straightforward because the “biological age” of the tumor (i.e., the time period over which cancer cells have been accumulating mutations) does not necessarily correlate with the biological age of the patient. For instance, a geriatric patient could have developed a rapidly growing tumor over a period of months while a young adult could have had an indolent tumor since childhood. To attempt this proposed validation, at minimum we would need to obtain additional clinical information to best estimate the time of tumor initiation. Collecting such information goes beyond the scope of our current manuscript, which includes more than 2,000 samples.

This section is not particularly novel as it validates previously published datasets (PMID: 32042194 and 33016928). Concern that the methods used do not provide adequate resolution to accurately generate mutational signatures.

As the reviewer mentions, our results and our main conclusions from this analysis largely coincide with those previously reported in the studies the reviewer references, which supports the validity of our results. Further, the justification and prior use of the same threshold for mutational signature analyses in a variety of previous studies (e.g. Zehir et al. Nat Med 2017, Caso et al. J Thorac Oncol 2020, Jones et al., Clin Cancer Res 2021) indicate that our methodology is adequate. Lastly, the two studies referenced by the reviewer are specific to one sarcoma subtype, angiosarcoma, while our analysis includes many additional subtypes, which may be of interest to readers.

11. Although the panel can provide data on MSI status, it would be appropriate to justify why this would be informative in sarcoma, as in this dataset and others, MSI is not considered a major DNA damage pathway in this disease. Was there any data to suggest NHEJ dysfunction or HRD? Why not use HRD detect (PMID: 33283135)? It is critical to develop classifiers and tools that can be used for discovery and/or inform mesenchymal biology. Suggest Figure 6 should be supplemental.

We thank the reviewer for raising this important point. While prior studies such as the TCGA indicate that sarcomas are in general microsatellite-stable, given the importance of MSI as a predictive biomarker for response to immune checkpoint blockade, we decided to report MSI status across our cohort because it includes 45 sarcoma subtypes compared to the 7 included in the TCGA. While we did not identify any subtypes that were significant outliers, we respectfully suggest that investigating this question and providing the analysis for the community is worthwhile. However, we agree that Figure 6 would be more appropriately presented as supplemental and have revised the manuscript accordingly.

With regard to the suggestion of applying an HRD score as we have done previously, we did not perform that analysis here because HRD score in our prior study was not associated with progression-free or overall survival. However, by making the underlying data available to the community, investigators with a special interest in DDR will be able to use the same dataset to assign HRD scores and apply other methodologies of interest.

We also agree that it is important to develop novel tools for discovery in mesenchymal biology, but the design and testing of novel methods is beyond the scope of this study.

Additionally, the following minor comments should be addressed:

1. Line 16 states that sarcomas arise in “organ sites”. Most sarcomas do not arise in organs per se, please clarify and suggest coupling with phrase “connective tissue”. Also, in Figure 1A please define the nomenclature of retroperitoneal vs. retro/IA? How does this distinction help in prognostication?

We agree with the reviewer and have changed the text in line 17 to read “*anatomic locations and connective tissue types*”. We have also added the definition of “retro/IA” to the legend for Figure 1.

2. In Line 18 please provide context for studying pediatric sarcomas, as they are in fact a common solid tumor in this age group. Does this manuscript seek to address molecular

signatures across age span? Also, TCGA, which is foundational for the panel design didn't evaluate pediatric sarcoma subtypes. Does this impact the utility of sequencing of pediatric subtypes that MSK-Impact is now being applied to?

We included pediatric cases in our study for completeness because some of the histotypes that we analyzed occur mostly in patients of very young age. The utility of sequencing pediatric tumors with MSK-IMPACT has been discussed and demonstrated in a separate manuscript reporting a germline mutational analysis (Fiala et al. *Nat Cancer* 2021). To clarify the elevated frequencies of sarcomas in young patients, we have added the following language and reference to lines 18–20, "*Sarcomas are also rare tumors, representing < 1% of all malignancies in adults, though more common in the pediatric population where they represent approximately 20% of non-hematologic malignancies (Burningham et al. Clin Sarcoma Res 2012)*".

3. The authors are to be commended for the detailed clinicopathologic data, especially survival outcomes provided in Figure 1. However, in some subtypes (ANGS) tumor purity is <50% in the majority of samples. Also, with respect to EDI, race does not appear to be a representative demographic of the US population, does this limit the interpretation and translatability?

We thank the reviewer for raising these points and have added mention both of these important points in a new paragraph summarizing the study's limitations in the Discussion (lines 450–452).

4. In Figure 2, it is of concern that the pathognomonic fusion for ARMS isn't found in the majority of specimens (? 30-40%), please clarify this why this wasn't the case. In the methods section it is commented that if there was a discrepancy in diagnosis this was reviewed by a sarcoma expert and reassigned. Please provide this data as would be informative in how this workflow occurs in a precision medicine program.

We thank the reviewer for raising this important question as we agree that we expect specific fusion events in ARMS. In Figure 2A, lack of assignment to the "Fusion (other)" category does not imply the lack of a pathognomonic fusion. Alterations were assigned to this category if the fusion was detected by a non-IMPACT method (e.g. RNA-seq or FISH). Fusions could also have been detected by IMPACT, in which case they were classified as either 1+ driver (IMPACT) or VUS only depending on oncogenicity as determined by OncoKB. This is stated in the Figure 2 legend: "Oncogenic fusions detected by MSK-IMPACT are classified as drivers." To provide additional clarity for the reader, we have added a new Supplementary Table 6, which lists all fusions and indicates the method by which each was identified. As demonstrated in the new supplementary table and Figure 2, in the case of ARMS, 25 of 29 samples have a *FOXO1* rearrangement.

We have also updated the Methods with additional detail on the expert review of a subset of these cases (lines 490–496) and specified the number of cases in which diagnoses were updated (91 of 2,138; < 5% of the cohort) in the Results (lines 52–54). Many of these updated diagnoses were to specify a specific subtype in cases where the original annotated diagnosis was a class of sarcomas with the subtype not specified (e.g. alveolar RMS to replace RMS).

5. Please clarify nomenclature of ULMS vs LMS this is confusing at times. Please provide data in Figure 5A for LMS as only ULMS is represented.

Uterine leiomyosarcoma is abbreviated as ULMS and soft tissue leiomyosarcoma is abbreviated as LMS following the Oncotree ontology nomenclature system (Kundra et al. JCO Clin Cancer Inform 2021) as referenced in the Methods (lines 501–504). The same Oncotree ontology classification approach was used in the first MSK-IMPACT publication (Zehir et al. Nat Med 2017). We acknowledge that in a study with 45 histologic subtypes, the sheer number of Oncotree classifications can become cumbersome. For this reason, in Figure 1A we included a list of the most frequent subtypes along with their Oncotree classifier abbreviation as an aid to the reader.

We did not include LMS in Figure 5A as there was no statistically significant co-occurrence or mutual exclusivity identified in that group of tumors. The legend for Figure 5A has been revised to clarify this point.

6. Supplemental Figure 1B label needs to be corrected.

This has been corrected. We thank the reviewer for bringing this to our attention.

Reviewer #2 (Remarks to the Author): Expert in bioinformatics, sarcoma genomics and subtypes

Singer et al. propose a unique pan-sarcoma mutational landscape of 22 sarcoma types based on 1918 samples profiled on the MSK-IMPACT targeted gene panels.

They first describe the cohort and cohort-level clinical features, including purity, sequencing depth, age and survival; then, the number and type of driver alterations detected, pointing to clustered FLT4 VUS missense mutations in angiosarcomas; then copy-number aberrations (CNA), the prevalence of CNA events and whole genome doubling (WGD), relating WGD status to survival in metastasis of UPS; gene-level and pathway-level mutation frequencies across sarcoma types, with interesting high-prevalence PI3K, TERT and epigenetic-related alterations highlighted; they look at gene-level and pathway-level co-occurrence and mutual exclusivity of alterations, with a detailed look at the common ATRX mutations; and propose a summary view across types by clustering samples by their alteration profiles; they go on to characterise MSI and TMB, two clinically relevant features, with mutational signature analysis showing UV signature in high TMB samples; and finally look at the clinical actionability of the mutations.

Their data release accompanying the manuscript would also include data for an additional 220 samples for other underrepresented sarcoma types, not analysed here.

This is an important resource for the field, first of this kind in sarcoma genomics. The manuscript is rich in details and the data is well-presented and well-described. It offers interesting parallels and distinctions across these 22 different sarcoma types and a "global view" based on their genomic alterations as well as pointers for clinical management.

We thank reviewer #2 for recognizing the value of this manuscript as an important resource for the field. We hope that the analysis herein will inform future studies by the sarcoma community that will ultimately improve the care of sarcoma patients.

We have a few comments that we hope could improve this manuscript.

Comments:

* It is important that this dataset is released with the manuscript. The data has not been made available to reviewers but there is a promise of "bulk download" of genomic and clinical data through the cBioPortal website. What data and in which format it will be available could be made clearer.

This is a very legitimate point. To provide full transparency about the data that we will release, we have enabled public access to our cBioPortal study, so that reviewers can access it and browse through it using the following link:

https://www.cbioportal.org/study/summary?id=sarcoma_mskcc_2022

Reviewers and future readers of the manuscript can use the following link to download all clinical and genomic data used in our analyses:

https://cbioportal-datahub.s3.amazonaws.com/sarcoma_mskcc_2022.tar.gz

This link points to a compressed file that contains the following files:

- data_mutations_extended.txt (Mutation Annotation Format (MAF) file containing mutation calls for all somatic variants in our dataset
- data_CNA.txt : file containing gene-level copy number calls
- data_cna_hg19.seg : segmentation file containing copy number information
- data_fusions.txt : file containing detailed information about all the mutations called based on sequencing data from the MSK-IMPACT panel
- data_clinical_patient : clinical information at the patient level
- data_clinical_sample: clinical information at the sample level

These links have been added to the Data Availability section of the Methods.

* Metastases and primary samples are used as a combined group. But are there differences between metastases and primary samples?

We agree that examining differences between metastatic and primary samples could be informative and hypothesis-generating. In specific instances, such as in the context of WGD outcomes analysis (Figure 3D and Supplementary Figure 1 B,C), we did compare metastatic and primary groups. We also investigated this question in detail in a pan-cancer manner in a separate manuscript that was recently accepted for publication in *Cell* (pre-print available at <https://www.biorxiv.org/content/10.1101/2021.06.28.450217v1>). In that study, we investigated genomic differences between primary tumors and metastases in some of the histotypes in our

current sarcoma manuscript that had sufficiently large numbers of samples to allow meaningful analyses (GIST, UPS, LMS, and some liposarcomas). In GIST, we identified an increase in FGA and TMB in metastatic samples vs. primary samples, as well as an increase in the frequency of MYC pathway alterations. In UPS, we reported a decrease in the frequency of *JUN* amplification events in metastatic samples. We did not observe any other significant differences between primaries and metastases at the genomic level. Because these analyses are reported elsewhere, we did not consider it appropriate to repeat them here and, instead, we have added the following sentence to the limitations paragraph of the Discussion (lines 441–447) to point interested readers to the other manuscript:

“For most of our analyses, we combined primary and metastatic samples. This choice is supported by a detailed comparison of primary vs. metastatic samples for the more common histotypes, which did not reveal any major significant differences at the genomic level, with the exception of increased TMB and FGA in metastatic vs. primary GIST samples and fewer JUN amplifications in metastatic vs. primary UPS samples

[<https://www.biorxiv.org/content/10.1101/2021.06.28.450217v1>].”

* As the authors are using data from different versions of their gene panel (with increasing number of genes), it is not clear how frequencies and counts are derived? I think absolute counts should be shown, while frequencies should be based on the number of samples for which the gene is included in the panel (not total amount of samples). Is this the case? If so, a brief description of how this is taken into account in the methods is missing.

We thank the reviewer for raising this important question. When alterations are viewed using the cBioPortal, frequencies are automatically adjusted to account for different gene panels by dividing the number of samples for which the gene was altered by the number of samples for which that specific gene was sequenced. We have followed the same approach to compute all frequencies reported in our manuscript. To make this clear, and following the reviewer’s suggestion, we have added the following sentence to our Methods section (lines 527–529):

“All reported alteration frequencies were adjusted to account for the specific set of genes included in each version of the MSK-IMPACT panel by dividing by the number of samples for which a given gene was sequenced.”

* The variant calling strategies are not described in the methods. How were variants called? Please provide at least a link to a previous paper where this is described to make the analyses interpretable/reproducible.

We apologize for this omission in our previous version of the manuscript and agree this is an important point. We have added the following text to the Methods (lines 508–521) to provide technical details about the MSK-IMPACT sequencing and variant calling pipelines, including references to two previous papers and a Github repository where interested readers can find full details regarding our variant calling pipeline:

“Targeted DNA Sequencing using MSK-IMPACT

Sequencing was performed using MSK-IMPACT, a hybridization capture-based next-generation sequencing assay (Cheng et al. J Mol Diagn 2015, in a Clinical Laboratory Improvement Amendments (CLIA)-certified molecular laboratory. Genomic DNA from formalin-fixed paraffin-embedded (FFPE) primary or metastatic sarcomas and patient-matched normal blood samples

was extracted and sheared, and custom probes were synthesized for targeted sequencing of all exons and selected introns of 341, 410, or 468 genes as previously described (Cheng et al. J Mol Diagn 2015, Zehir et al. Nat Med 2017). Pooled libraries containing captured DNA fragments were sequenced using the Illumina HiSeq 2500 to high, uniform coverage (>500× median coverage). All classes of genomic alterations including substitutions, indels, copy number alterations, and rearrangements were determined and called against the patient's matched normal sample. The computational pipelines used for variant calling are based on standard best practices using a combination of open-source and custom written scripts and programs, as previously published (Cheng et al. J Mol Diagn 2015, Zehir et al. Nat Med 2017). Custom code used for analyses is publicly accessible (<https://github.com/mskcc>)."

* In the methods, please describe exactly what data goes in the UMAP+HDBSCAN. This is one of the main pan-sarcoma results, but it is not clear how it was derived.

The input to the UMAP-HDBSCAN modules was a binary matrix of oncogenic events, where each row corresponds to a sample and each column corresponds to a different gene. The entry corresponding to column x and row y is equal to 1 if the xth gene has an oncogenic mutation, copy number alteration or fusion in the sample sequenced for the yth patient, and equal to 0 otherwise. This is the standard Genomic Alteration Matrix (GAM) that we have used in previous studies and publications (e.g., Sanchez Vega et al. Cell 2018). We have updated the relevant portion of the Methods (lines 552–561) to incorporate this information such that it now reads:

"Clustering analysis was performed as follows. All mutations, fusions, and copy number alterations were filtered for functional relevance using OncoKB. These oncogenic alterations were then aggregated into binary matrix format per gene for each patient and filtered using the 341-gene list on the IMPACT panel to generate the clustering input. Input matrix dimensionality was reduced using Uniform Manifold Approximation and Projection (UMAP) (<http://arxiv.org/abs/1802.03426>) via the R package umap. Clustering was performed using the Hierarchical Density-Based Spatial Clustering of Applications with Noise (HDBSCAN) method (Campello, Moulavi, Sander PAKDD 2013: Advances in Knowledge Discovery and Data Mining) via the R package dbscan (Hahsler, Piekenbrock, Doran. J Stat Software 2019). All samples labelled NA (cluster 0) were unassignable to a cluster. Shannon entropy was calculated from observed cluster assignment by subtype and reported in natural units."

* In silico ploidy inference (and thus WGD status) is underdetermined and ambiguous (discussed here:<https://doi.org/10.1038/s41592-020-01013-2>), and in the presented study, WGD prevalence seems underestimated at least for UPS. Indeed, 65% TCGA samples underwent WGD (<https://doi.org/10.1016/j.cell.2017.10.014>); in Steele et al., 90% of undifferentiated sarcomas showed WGD with experimental validation of ploidies (<https://doi.org/10.1016/j.ccell.2019.02.002>). We would expect even higher rates in the dataset of Singer et al., as >30% of cases are metastatic samples, expected to present with even higher prevalence of WGD. Because of the limited genome resolution of these targeted panels, perhaps FACETS would favour low-ploidy solutions; in any case, the ploidy and WGD status should be interpreted with a grain of salt, i.e. an appropriate discussion would be useful.

In a previous study from several members of our team, we showed that WGD estimates from MSK-IMPACT have very good concordance with estimates derived from WES (Bielski et al., Nat Genet 2018). In particular, we evaluated matched MSK-IMPACT and WES data for 149 patients

and found the WGD calls to be concordant in 147 of them (99%), which confirmed the robustness of WGD inference using targeted sequencing data. Still, because other publications may have used different analytical approaches and different sequencing platforms to estimate WGD status, it is not surprising to observe discrepancies in reported frequencies. We have added the following text to the Discussion (lines 439–441) to explicitly mention this fact:

“In addition, while WGD estimates from MSK-IMPACT are generally concordant with those derived from whole exome sequencing (Bielski et al., Nat Genet 2018), the WGD frequencies reported herein may be discordant from those derived by alternative analytical methods and sequencing platforms.”

* In general, the pros and cons of the targeted panel approach could be discussed better in the context of this pan-sarcoma genomics study. The identification of important genetic changes, i.e. focal CNA, fusions, ploidy changes/WGD status, and unknown driver genes is naturally limited. Pan-sarcoma whole-genome sequencing studies are needed to identify more events that are not captured here. Though this may be trivial, this should be mentioned and discussed by the authors.

We agree with the reviewer that although targeted sequencing studies provide significant value, they are not designed to capture all pathologic events that occur in cancer genomes. We have therefore added a paragraph to the Discussion (lines 432–441) in which we discuss the inherent limits of targeted sequencing and, as the reviewer suggests, propose that additional studies including whole genome sequencing will be important for adding additional depth to our understanding of sarcoma genomics.

* The rationale behind the selection of a single representative sample in the multi-sample cases (= early date, purity, coverage) is not really explained. Why does it make sense in that order? Instead why not e.g. pick primary site over metastasis, then the highest number of reads per tumor chromosomal copies (compound power metric based on copy-number, read depth and purity, please see: <https://doi.org/10.1016/j.cell.2021.03.009>, <https://doi.org/10.1038/s41592-020-01013-2>), then select the largest gene panel (=widest genomic coverage)?

We thank the reviewer for raising this important question. Our primary goal in selecting a single sample per patient was to prevent overrepresentation of particular genetic events given the presumed clonal relationship between samples arising from the same primary tumor. There are multiple reasonable paths to achieve this important goal. Our rationale for prioritizing the sample with the earliest date was to reduce the potential influence of treatment-induced genetic changes, because earlier samples are more likely to be collected from patients who have received fewer lines of therapy. This criterion alone was sufficient to select a sample in the vast majority of cases, as sequencing of multiple samples from a single patient collected on the same date was rare. Still, for completeness, we decided to choose the highest purity sample to disambiguate the choice in those very rare cases and, in the extremely rare event of two samples being sequenced on the same date and having the same inferred purity, we chose the one with the highest average depth of sequencing. While alternative approaches, such as the one suggested by the reviewer, have inherent benefits, they would also have certain drawbacks. For instance, using a larger gene panel size as a criterion would favor including samples collected at later timepoints because the gene panel expanded over time. This could introduce a confounder of additional treatment-related events into the analysis. We want to emphasize the

fact that > 85% of the patients in our cohort had only one sequenced sample, so a different protocol for selecting a representative sample (e.g., prioritizing primaries over metastases instead of the earlier sequenced sample) would affect <15% of the cohort, while many selections even within that small fraction would remain unchanged, and are therefore unlikely to significantly alter our reported results and conclusions.

To explain the rationale for our method of sample prioritization, we have added the following sentence to the Methods section (lines 481–484):

“Earlier sequenced samples were prioritized to reduce the potential influence of treatment-induced genetic changes because these are more likely to have been collected from patients who received fewer lines of therapy.”

* This paper is a unique pan-sarcoma genomics paper, as it covers so many sarcoma types. But therefore, the underrepresented sarcoma types (220 samples, an extra 11%) should be included in the analyses. It is a big loss not to have them shown here. Especially, in the summary UMAP results, it might show where these other sarcoma types sit relative to the 22 analysed. If no strong signal comes out, they could be pooled in a category "Others" for most of the other figures.

We thank the reviewer for recognizing one of the major strengths of our manuscript, which is its inclusion of the rarest subtypes. For much of the analysis, we focused on subtypes represented by ≥ 20 samples to avoid biasing frequency calculations by including potential outlier events in groups with very small samples sizes. However, acknowledging the benefit of including rare subtypes for generating hypotheses, we did use a threshold of 10 samples per subtype in reporting epigenetic pathway events (Figure 4D and Supplementary Figures 2 and 3). This led us to highlight in the text (lines 224–227) the observation that uterine adenocarcinomas (UAS; $n=14$) have a high rate of SWI/SNF pathway alterations, which was not previously known. In addition, by basing our figure for *TERT* alterations (Figure 4C) on the 9 most altered subtypes, we include intimal sarcomas (INTS, $n=18$), which is another rare subtype, which we highlight in the text (lines 177–181) given that this event had not been previously reported.

With regard to including all subtypes in the UMAP clustering, we very much appreciate this excellent suggestion and have updated the analysis accordingly in a new Figure 5C-D and have updated the Results (lines 295–315) and Discussion (lines 457–468) accordingly to discuss several rare subtypes now included in the analysis.

We make a final point based on our agreement that genomic characterization of the rarest subtypes is an extremely valuable resource to the field, which is that we will include *all samples* in our data release on the cBioPortal platform for further analysis by other investigators. It is our hope that by making these data available, we will help to advance global efforts to understand the underlying biology of sarcomas and improve their treatments.

* Figure 1: avoid loaded term of "Race"? Perhaps replace by "Population"?

We will work with the editorial staff at *Nature Communications* to adhere to publisher style guidelines and/or additional guidance regarding the use of the term “race” if the manuscript is accepted for publication.

* Supp Fig 4B: x-axis labels read "Copy number slteration"

We have corrected this typographical error and thank the reviewer for bringing it to our attention.

REVIEWER COMMENTS

Reviewer #1 (Remarks to the Author):

The reviewer thanks the authors for responsive feedback to the major and minor queries on this manuscript. They are to be congratulated for the quality and impact of this manuscript.

One following small issue warrants correction prior to publication:

Supplemental Figure 4B, the p values are not clear and graphics need to be cleaned up.

Reviewer #2 (Remarks to the Author):

The manuscript has significantly improved and the authors have satisfyingly answered all my comments.

I thank the authors for their thorough consideration of my comments and wish to congratulate them on this piece of work.

REVIEWERS' COMMENTS

Reviewer #1 (Remarks to the Author):

The reviewer thanks the authors for responsive feedback to the major and minor queries on this manuscript. They are to be congratulated for the quality and impact of this manuscript.

We agree that the manuscript has been improved by the reviewers' input and thank them for their valuable suggestions.

One following small issue warrants correction prior to publication:

Supplemental Figure 4B, the p values are not clear and graphics need to be cleaned up.

The formatting issue in the referenced supplementary figure has been corrected.

Reviewer #2 (Remarks to the Author):

The manuscript has significantly improved and the authors have satisfyingly answered all my comments.

I thank the authors for their thorough consideration of my comments and wish to congratulate them on this piece of work.

We agree that the manuscript has been improved by the reviewers' input and thank them for their valuable suggestions.